# Navigation-grade interferometric air-core antiresonant fibre optic gyroscope with enhanced thermal stability

Maochun Li [1,7], Yizhi Sun [2,3,4,7], Shoufei Gao[2,3,4,7], Xiaoming Zhao [1] ✉, Fei Hui[1], Wei Luo[1], Qingbo Hu[2,3], Hao Chen [2,3], Helin Wu[2,3], Yingying Wang[2,3,4], Miao Yan[1,5] & Wei Ding [2,3,4,6] ✉

We present a groundbreaking navigation-grade interferometric air-core fibre optic gyroscope (IFOG) using a quadrupolar-wound coil of four-tube truncated double nested antiresonant nodeless fibre (tDNANF). This state-of-the-art tDNANF simultaneously achieves low loss, low bend loss, single-spatial-mode operation, and exceptional linear polarization purity over a broad wavelength range. Our 469 m tDNANF coil demonstrated a polarization extinction ratio (PER) of ~20 dB when illuminated by an amplified spontaneous emission (ASE) source spanning 1525-1565 nm. Under these conditions, the gyro archives an angular random walk (ARW) of 0.00383 deg h$^{-1/2}$ and a bias instability (BI) drift of 0.0017 deg h$^{-1}$, marking the first instance of navigation-grade performance in air-core FOGs. Additionally, we validated the low thermal sensitivity of air-core FOGs, with reductions of 9.24/10.68/6.82 compared to that of conventional polarization-maintaining solid-core FOGs of the same size across various temperature ranges. These results represent a significant step towards long-standing promise of high-precision inertial navigation applications with superior environmental adaptability.

The fibre optic gyroscope (FOG), which relies on the Sagnac effect, is one of the most successful optical fibre sensors and serves as the core equipment for inertial navigation, positioning, and attitude determination[1–3]. Due to their high resolution and simple structure, often referred to as the minimum reciprocal scheme[1], closed-loop interferometric fibre optic gyroscopes (IFOGs) have been widely employed in both military and civilian fields, including aviation, aerospace, weapon systems, autonomous vehicles, oil platforms, and well logging, often preferred over ring laser gyroscopes[4,5] and microelectromechanical system gyroscopes[6]. The comprehensive capabilities of the IFOG make it the primary candidate for inertial navigation systems both currently and in the foreseeable future, necessitating further optimization.

The prime performance of a FOG can be analysed from three aspects: short-term noise (e.g., angular random walk, ARW), long-term bias drift (e.g., bias stability/bias instability, BS/BI), and environmental adaptability, as outlined in Fig. 1. Decades of technological efforts to improve all FOG components, including the optical source, modulation strategy, coil winding, and detector, have pushed the IFOG to the pinnacle of these performance metrics[7]. However, the physical attributes of the fibre medium impose fundamental limits on further improvement of IFOGs. In conventional polarization-maintaining silica core fibres (SCFs), several deleterious effects inherently exist for gyro,

[1]Tianjin Key Laboratory of Quantum Precision Measurement Technology, Tianjin Navigation Instruments Research Institute, Tianjin 300131, China. [2]Guangdong Provincial Key Laboratory of Optical Fibre Sensing and Communication, Institute of Photonics Technology, Jinan University, Guangzhou 510632, China. [3]College of Physics & Optoelectronic Engineering, Jinan University, Guangzhou 510632, China. [4]Linfiber Technology (Nantong) Co. Ltd., Nantong, Jiangsu 226010, China. [5]School of Mechanical Engineering, Nanjing University of Science and Technology, Nanjing 210094, China. [6]Pengcheng Laboratory, Shenzhen 518055, China. [7]These authors contributed equally: Maochun Li, Yizhi Sun, Shoufei Gao. ✉e-mail: tjhhyq@yeah.net; dingwei@jnu.edu.cn

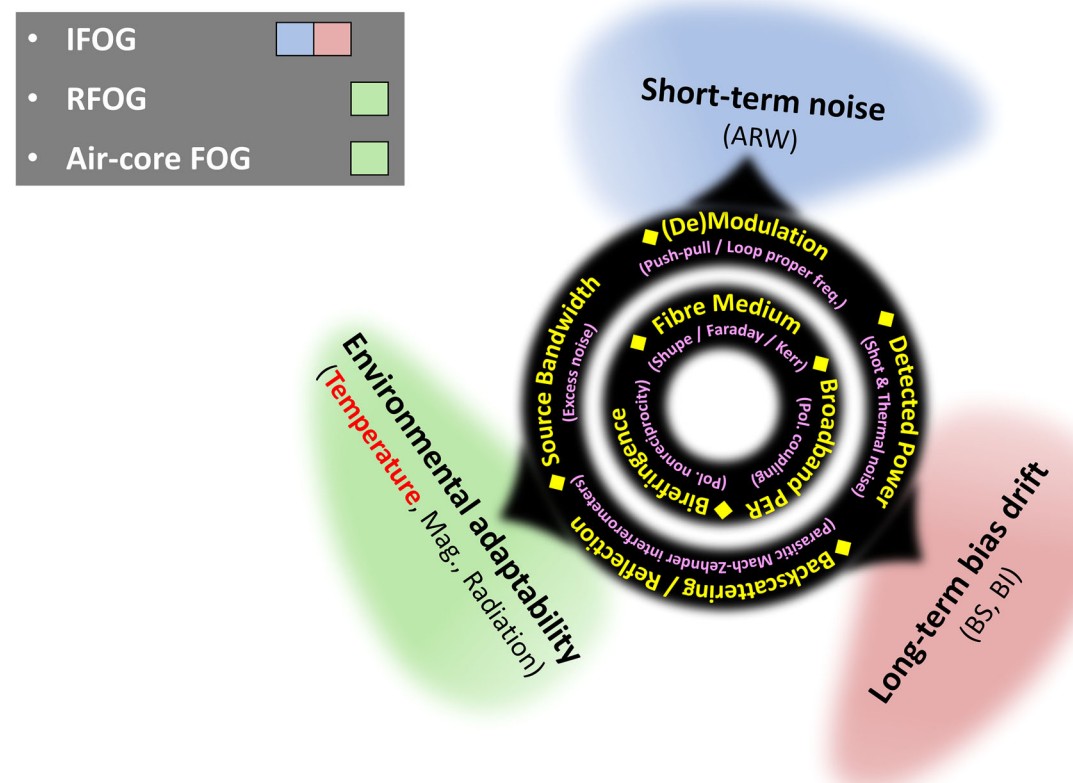

**Fig. 1 | Key metrics (yellow diamonds) of a fibre optic gyroscope (FOG) assessed from three performance aspects (labelled in blue, red, and green, respectively).** While practical interferometric fibre optic gyroscopes (IFOGs) have almost reached ultimate performances in short-term noise (e.g., angle random walk, ARW) and long-term bias drift (e.g., bias instability (BI) and bias stability (BS)), further improvements in environmental adaptability—including tolerances to variations of temperature, magnetic field, and radiation—are still under intense investigations, with possible solutions being explored by means of resonant fibre optic gyroscopes (RFOGs) and air-core FOGs. The primary challenge lies in circumventing all the deleterious effects (described in parentheses and in magenta) concurrently.

such as the temperature and strain induced time-transience effects[8,9], magnetic field Faraday effect[10], optical nonlinearity Kerr effect[11], Rayleigh backscattering[12], and radiation sensitivity[13]. High-precision IFOGs usually employ kilometer-level fibre coils, where the accumulation of these components amplifies the environmental influence. A possible solution is to utilize a resonant fibre optic gyroscope (RFOG) with a shorter fibre length[14]. However, the accuracy of RFOG has only recently reached navigation grade[15] and still markedly lags behind that of IFOGs.

Another promising approach to address environmental issues is to replace the fibre medium of FOGs from silica with air. Exploration in this direction dates back to the year 2006[16]. This change of the fibre medium dramatically reduced errors caused by the Faraday effect (by more than 20 times) and the Kerr effect (by more than 170 times)[17]. Initial research[18] predicted the thermal sensitivity of air-core FOGs to be 3-11 times smaller than that of SCF FOGs, and later experiments revealed a 6.5-fold reduction[19] with photonic bandgap-type hollow core fibres (PBG-HCFs)[20]

In the past decade, another type of HCF, referred to as anti-resonant HCFs (ARFs)[21,22], has made significant advances in attenuation reduction[23–26] and effective single-mode operation[27], facilitating the propagation length to reach the kilometer level and mitigating multi-path interference, which severely impairs the noise and drift performance of PBG-HCF-based FOGs[16,17,19]. Furthermore, recent experiments have validated that ARFs have better light transmission attributes than PBG-HCFs, such as an almost four orders of magnitude lower nonlinear coefficient[28] and Brillouin gain coefficient[29] than standard single-mode fibres (SSMFs), a lower backscattering coefficient (28 dB below that of SSMF[30]), virtually no radiation-induced attenuation[31], and up to 20-fold

reduction in thermal phase sensitivity compared to SSMFs[32]. ARFs, therefore, simultaneously possess advantages in environmental adaptability and modal purity compared to conventional SCFs and PBG-HCFs, respectively, implying their great potential for use in high-performance FOGs.

However, thus far, ARF-based FOGs have only been tested in RFOG configurations, with implementations of either a 5.6 m Kagome fibre ring[33] or a 136 m nested antiresonant nodeless fibre (NANF) resonator[34]. The observed bias stabilities, albeit greatly superior to those of PBG-HCF-based FOGs, remain at 0.15 deg h$^{-1}$ and 0.05 deg h$^{-1}$, respectively, which are still inferior to the practically usable levels.

The key to unlocking the full potential of ARF-based FOGs may lie in utilizing broadband sources[35], thereby obtaining the advantages of IFOG and maintaining polarization[36]. For the latter, the polarization behaviors in an ARF have two features. On the one hand, ARFs can maintain exceptional polarization purity by overcoming the fundamental anisotropic Rayleigh scattering limit[37] in SCFs. In ARFs, the polarization crosstalk primarily originates from the rough interfaces between air and silica rather than from the air core. The interpolarization power coupling coefficient can be more than three orders of magnitude lower than that in SCFs[38]. On the other hand, state-of-the-art ultralow-loss ARFs are usually non-polarization-maintaining. The principal axis angles of these ARFs have apparent wavelength dependence and are highly sensitive to bending and twisting, meaning that the linear polarization state can only be well preserved within a small wavelength range[38]. As a result, it is vital to balance the broadband light source and linear polarization maintenance in an IFOG of a small-radius coil.

In this work, a low-loss, low-bend-loss four-tube truncated double nested antiresonant nodeless hollow core fibre (tDNANF)[39,40] was

designed and fabricated. The modest $10^{-6}$ level of birefringence and the principle-axis-angle-holding capability, inherently facilitated by its geometry, led to an approximately 20 dB thermally stable polarization extinction ratio (PER) in a 469 m quadrupolar-wounded fibre coil over the wavelength range of 1525-1565 nm. The combination of all beneficial factors—a quadrupolar-wound coil of sufficient fibre length, simple fibre-to-chip connections, a high spatial mode purity, and a high linear polarization purity—in such an IFOG yields an ARW at a high power of 0.00383 deg h$^{-1/2}$, a BS drift over 100 s of 0.023 deg h$^{-1}$, and a BI drift of 0.0017 deg h$^{-1}$, representing, to the best of our knowledge, the first navigation-grade air-core FOG. Furthermore, a decrease in thermal sensitivity by factors of 9.24/10.68/6.82, compared to that of a conventional polarization-maintaining SCF-FOG with the same coil length and diameter, was validated when our four-tube $t$DNANF-FOG was subjected to different temperature ramps. This exhibited better thermal stability than all the reported PBG-HCF-based FOGs[17,19].

## Results

### Four-tube truncated double nested antiresonant nodeless fibre structure

Designing an advanced HCF for high-precision FOGs requires a combination of low loss, low bend loss, single modality, and high linear polarization purity. These attributes should be maintained across a broad wavelength range, especially in the context of an IFOG configuration.

To reduce the loss of an ARF, introducing additional layers of glass membranes or nested tubes in the cladding area[39–41] is crucial. Recently, a DNANF has achieved a record loss of <0.11 dB km$^{-1}$ at 1550 nm, surpassing all other optical fibres[25]. Higher-order mode suppression, defined as the ratio of leakage losses of fundamental and higher-order modes, can be enhanced by designing cavities among the cladding tubes to selectively out-couple higher-order modes[27,42]. The DNANF structure is thus ideal for achieving low loss and single-mode operation. For generic ARFs, reducing bend loss involves using a small core with a diameter-to-wavelength ratio less than 20[22,23].

The main challenge in ARF design is maintaining high-purity linear polarization across a wide wavelength range. This can be achieved in a DNANF framework by adopting fourfold rotational symmetry. In ARFs, the intrinsic birefringence is dictated by nonuniformities among the core surrounding membranes and intertube gaps[43,44], with the former being more manageable. A variation in membrane thickness can create inconsistent antiresonant reflecting conditions. An ARF with fourfold rotational symmetry can efficiently transfer this inconsistency to high birefringence, especially when the core diameter is reduced, to increase the mode field overlap with the membranes[45], although this brings about higher loss. More importantly, if the principal axis direction is primarily dictated by a single segment of the core surround, its wavelength dependence can be greatly mitigated.

We designed and fabricated a four-tube $t$DNANF with a core diameter of 28.2 µm to facilitate low bend loss and high birefringence. The purpose of partly truncating the four outer tubes[46] is to shrink the void regions behind the intertube gaps since these cladding regions may create phase matches with the core fundamental mode. The membrane thicknesses of the outer tubes are measured to be 1.08–1.16 µm, corresponding to a normalized frequency (defined as $2t\sqrt{n^2-1}/\lambda$) of ~1.5 at 1550 nm. Here, $t$ is the membrane thickness, $n$ is the refractive index of glass, and $\lambda$ is the wavelength. A normalized frequency close to a half-integer indicates operation in the middle of an antiresonant transmission band, favouring low loss and wavelength independence.

We numerically calculated three realistic ARFs: one four-tube $t$DNANF, one five-tube NANF (both fabricated in-house), and one six-tube NANF reported in ref. 38. Scanning electron microscopy (SEM) images and structural parameters are presented in Fig. 2a and its caption. All the ARFs have nearly the same core diameters and small gap sizes. For the six-tube NANF (referred to as NANF-SI in ref. 38), no

membrane thickness variation is provided. Using finite element method simulation (see Methods), we obtained birefringence and relative principal axis angles in the 1510–1580 nm range (Fig. 2b). Without bending, the four-tube $t$DNANF exhibits birefringence almost an order of magnitude greater than that of the five-tube and six-tube NANFs. Additionally, the principal axis angle variation of the four-tube $t$DNANF (<0.4°) is much smaller than that of the five-tube NANF (<6°) and the six-tube NANF (~70°, experimental data) across 1525–1565 nm, which is the wavelength region of our IFOG light source (see Subsection Gyro test).

For FOG applications, it is crucial to consider the impacts of fibre bending on the birefringence and principal axis angle. In the simulation shown in Fig. 2c, with a bend radius of 6 cm (consistent with our IFOG, as detailed in Subsection Gyro test), the birefringence of the four-tube $t$DNANF remains an order of magnitude greater than that of the five-tube NANF. Despite the discontinuous rotational symmetry in fibre cross-sections affecting the polarization properties differently when bent at various orientations, the simulation (Fig. 2c) indicates that the principal axis angle offset of the four-tube $t$DNANF remains less than 2° across the 1525–1565 nm range. This may render a high PER when broadband linearly polarized light is launched into the fibre coil (see "Methods" and Supplementary Materials Section S1.3).

### Fibre optical characterization

To validate the analysis presented in the previous subsection, we characterized optical properties of our four-tube $t$DNANF, including propagation loss, macrobend loss, spatial mode purity, the strain-free Shupe constant, and linear polarization purity. The results are shown in Fig. 3.

Figure 3a shows the loss spectrum measured using the cut-back method from 510 m to 10 m, with the fibre wound on a drum with a radius ($R_b$) of 16 cm. This revealed an average propagation attenuation of 0.38 dB km$^{-1}$ within the wavelength range of 1525–1565 nm. To measure the bend loss, a section of the fibre was unwound from the drum and bent to a radius of $R_b$ = 6 cm for 100 turns. The transmission spectra recorded before and after bending indicate macrobend loss, depicted by the red line in Fig. 3a, with an average value of ~4.7 dB km$^{-1}$. Additionally, Fig. 3b shows the monitored loss during quadrupolar fibre winding[47] (see "Methods" and Supplementary Materials Section S3.1). A 40 nm bandwidth amplified spontaneous emission (ASE) source and a photodetector were assembled on and rotated with the two supply trays in our fibre winding setup. An overall extra loss of 1.51 dB was detected during winding when the 469 m fibre length was wound with 36 layers and 30 turns, featuring an inner radius of 6 cm and an outer radius of 7.6 cm. The linear fit in Fig. 3b indicates an additional loss of 3.3 dB km$^{-1}$, which is consistent with the measured macrobend loss at $R_b$ = 6 cm. After curing, the coil yielded a total loss of 4.23 dB, comprising 2.53 dB additional loss from potting adhesive curing, 1.51 dB additional loss from winding, and 0.19 dB fibre background loss.

To assess the spatial mode purity, a spatially and spectrally resolved imaging ($S^2$-imaging) method[48] was employed for the $t$DNANF coil at 1550 nm (see "Methods" and Supplementary Materials Section S1.1). As shown in Fig. 3c, no LP$_{11}$ or other higher-order modes are discernible above the noise floor (~−70 dB) at the output of the fibre coil. Resonant higher-order mode filtering[27], especially under bending state, facilitates such a high level of single-modedness, thus preventing intermodal interference in FOG operation.

To quantify the Shupe constant, defined as the *relative* phase change with temperature, $S = 1/\varphi \cdot d\varphi/dT$, where $\varphi$ is the phase accumulated along the fibre and $T$ is the temperature, an all-fibre Mach–Zehnder interferometer was employed. In our setup, a 10 m four-tube $t$DNANF coiled with an $R_b$ of 6 cm served as the signal arm and was placed inside a thermal chamber (see "Methods" and Supplementary Materials Section S1.2). As shown in Fig. 3d, the linear fit of

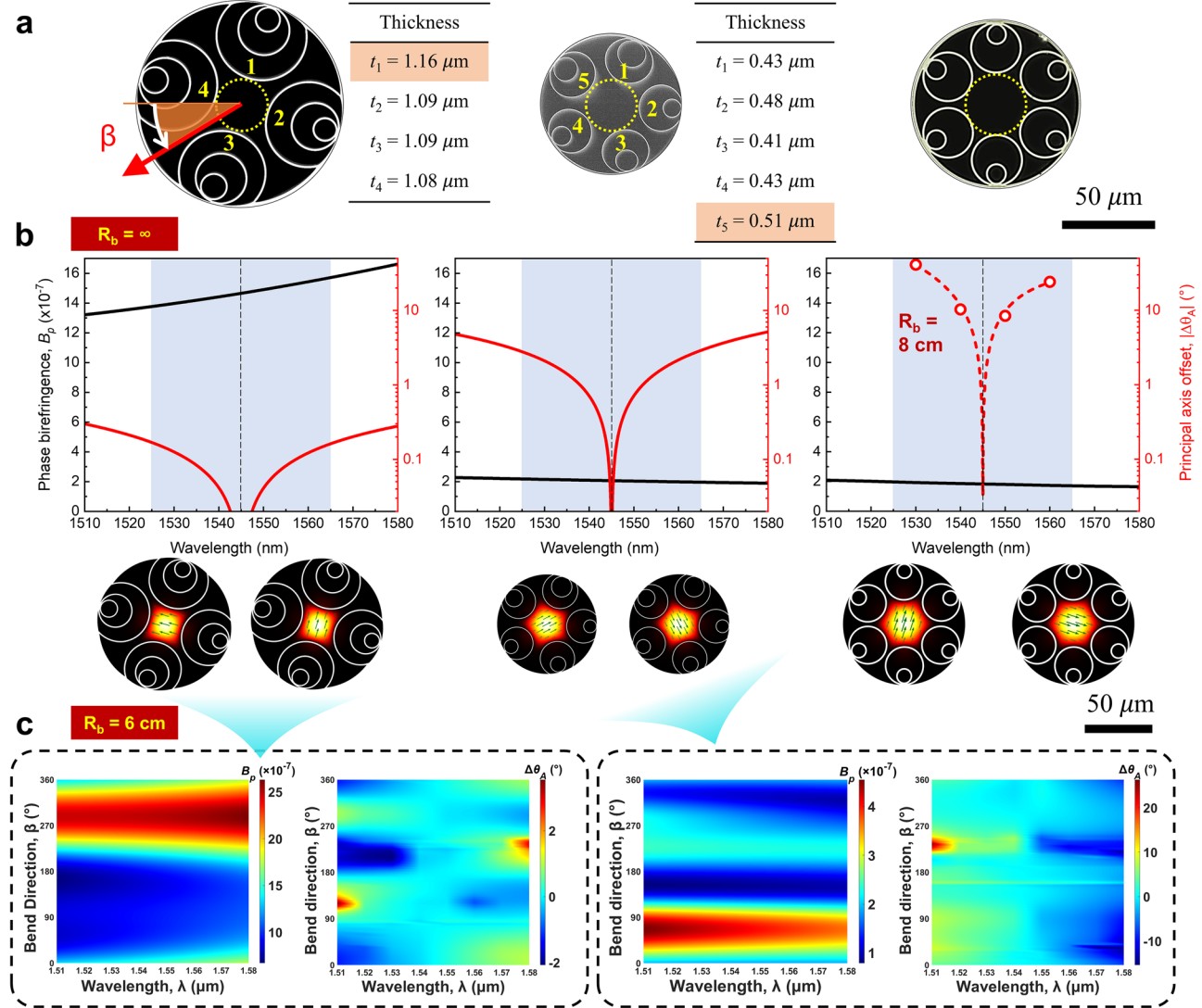

**Fig. 2 | Finite element modelling of polarization properties of three antiresonant fibres (ARFs). a** Scanning electron micrograph cross-sections of the four-tube truncated double nested antiresonant nodeless fibre ($t$DNANF) and the five-tube nested antiresonant nodeless fibre (NANF) fabricated in-house, alongside the six-tube NANF reported in ref. 38 (referred to as NANF-SI therein). The core diameters of the three ARFs are 28.2 μm, 27.9 μm, and 33.4 μm, respectively. The outer tube thicknesses are 1.08–1.16 μm, 0.41–0.51 μm, and 1.22 μm, respectively. The gap sizes are measured to be 5.4–6.2 μm, 5.3–7.1 μm, and 1.8–3.1 μm, respectively. **b** Simulated phase birefringence ($B_p$) and principal axis offsets ($\Delta\theta_A$, with respect to that at 1545 nm) with no bend and in the wavelength range of 1510–1580 nm. The structural asymmetry residual in realistic ARFs—e.g., the variation of the membrane thickness of the outer tubes, as outlined in panel **a**—gives rise to wavelength-dependent $B_p$ and $\Delta\theta_A$. For the six-tube NANF, the experimental results of $\theta_A$ under a bend radius ($R_b$) of 8 cm are acquired from Fig. S4 in ref. 38, and the simulation of $B_p$ utilizes the structural parameters listed in Table S1 therein together with an invariant membrane thickness of 1.22 μm. For all the three ARFs, the simulated mode-field profiles of the two polarizations at 1545 nm are plotted below with the arrows indicating the vector direction of the transverse electric field. **c** Simulated $B_p$ and $\Delta\theta_A$ of the four-tube $t$DNANF and the five-tube NANF as a function of wavelength ($\lambda$) and fibre bend direction ($\beta$) under $R_b$ = 6 cm. $\Delta\theta_A$ are defined as the angle offsets from the principal axis directions at 1545 nm. The red arrow in panel **a** depicts the bend direction from the fibre toward the centre of curvature.

the phase variation at 1550 nm yields $S$ = 0.52 ppm °C$^{-1}$, which is 14.4 times smaller than that of the control polarization-maintaining SCF (PMF, $S$ = 7.5 ppm °C$^{-1}$). Our measurement agrees well with the results in ref. 32, where the authors demonstrated that for a generic NANF, the thermal expansion of fused silica—being the dominant effect—contributes $S \approx 0.55$ ppm °C$^{-1}$ at room temperature. With a reduced Shupe constant, the rotation rate error of our four-tube $t$DNANF gyro, which is proportional to the product of the group and effective modal indices ($n_g \times n_{eff}$) and $S$[1,19], is expected to be ~30 times smaller than that of an SCF gyro in temperature regions where the second-order time derivative of temperature ($d^2T/dt^2$) is non-zero—conditions under which the *pure* Shupe effect dominates (refer to Fig. 6.5 in ref. 1).

However, it is widely recognized that in a real fibre coil, the Shupe constant is not uniform due to differential longitudinal strains. When

the coil is in the temperature regions with a non-zero first-order time derivative ($dT/dt \neq 0$), the Mohr effect—caused by layer-to-layer variation in the Shupe constant under changing temperature—becomes the primary source of thermal insensitivity[1,9]. To evaluate the impact of the Mohr effect, we calculate the fibre stiffness, defined as the product of Young's modulus and cross-sectional area[32]. Our four-tube $t$DNANF exhibits a 7.07-fold increase in stiffness compared to conventional PMF (see Supplementary Materials Section S1.4). Combined with the beneficial factor of $n_g \times n_{eff}$, the strain induced time-transient effect in our $t$DNANF gyro is expected to be ~14 times smaller than that of an SCF gyro. To further validate this reduction in the Mohr effect, future measurements with other coils will be needed.

A crucial metric of the IFOG configuration for evaluating the polarization-maintaining capability is the broadband PER, which

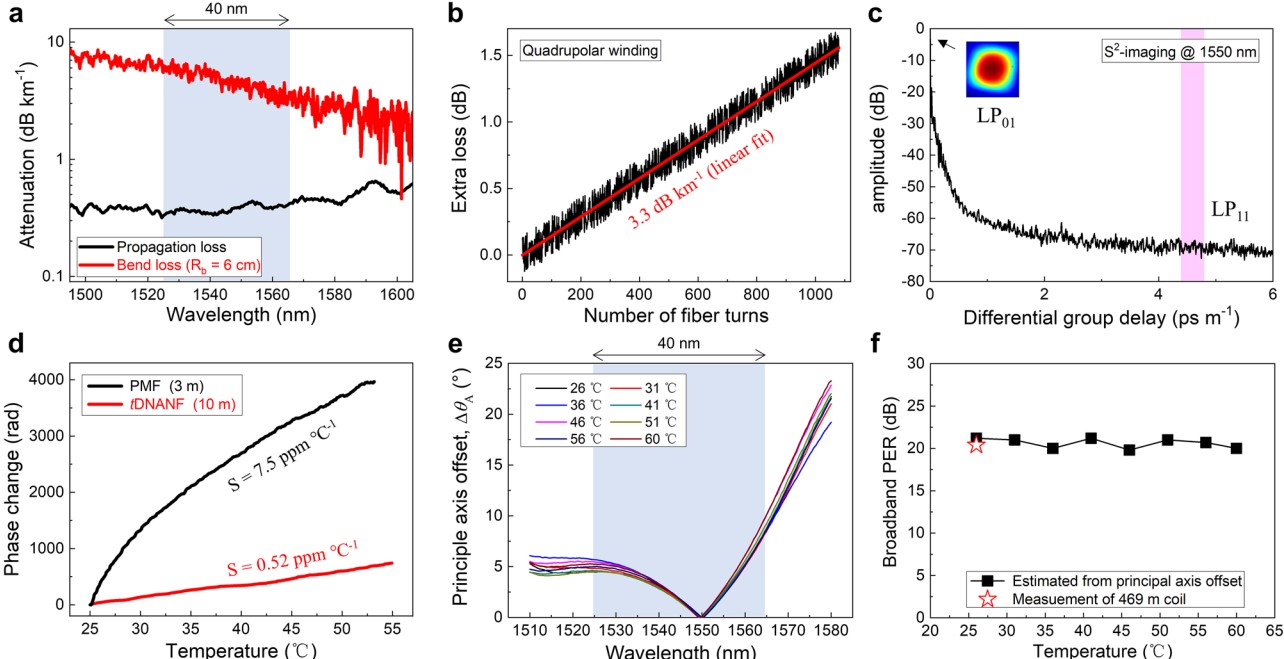

**Fig. 3 | Optical property characterizations of the four-tube truncated double nested antiresonant nodeless fibre (*t*DNANF).** **a** Measured spectral propagation loss and macrobend loss under a bend radius ($R_b$) of 6 cm. **b** Extra loss real-time recorded in a quadrupolar winding. The coil length is 469 m, an amplified spontaneous emission (ASE) source of 40 nm bandwidth is implemented, and the innermost and the outermost radii of the fibre coil are 6 cm and 7.6 cm, respectively. **c** Fourier transform of the optical transmission spectrum of the 469 m quadrupolar-wound coil at 1550 nm measured by spatially and spectrally resolved ($S^2$) imaging method. **d** Measured phase drifts over a 3 m polarization-maintaining fibre (PMF) and a 10 m four-tube *t*DNANF at 1550 nm with the temperature increasing from 25 to 55 °C, along with the fitted Shupe constants ($S$). **e** Measured principal axis offsets relative to the orientations at which minimum powers are detected in a crossed-polarizer setup at various temperatures. A 10 m four-tube *t*DNANF is measured in a 6 cm radius loop configuration. **f** Estimated extinction ratio between the two polarizations of the total integrated power from 1525 to 1565 nm based on the measured principal axis offset in (**e**) (black squares, see "Methods"), along with measured polarization extinction ratio (PER) of the 469 m coil by using an ASE source of 40 nm bandwidth and a commercial PER meter (red star).

measures how effectively a fibre (coil) preserves the linear polarization state of transmitted light over a specific wavelength range. The *ultimate* limit of broadband PER in an ARF is imposed by the wavelength dependence of the principal axis angle[38], which leads to the inevitable excitation of the two polarization states by a broadband linearly polarized light source (see Supplementary Materials Section S2). Based on the spectrally resolved Jones matrix model in ref. 38, the orientations of the principal axes at various wavelengths can be determined using a broadband cross-polarizer measurement setup (see "Methods" and Supplementary Materials Section S2.1). Limited by the resolution of our optical spectrum analyser (AQ6370D, Yokogawa), we measured a 10 m segment of four-tube *t*DNANF. As depicted in Fig. 3e, the offset of the principal axis angle ($\theta_A$) varies by approximately 10° within the wavelength range of 1525–1556 nm. This variation may be attributed to the structural asymmetry discussed in the previous subsection and the fibre bending with $R_b$ = 6 cm. Nevertheless, when the temperature changes from 26 to 60 °C, no significant variation in the principal axis angle spectrum is observed, which is consistent with ref. 38. The small temperature sensitivity of $\theta_A$ may result from changes in the microbending conditions, as the Young's modulus of our acrylate coating—which has a glass transition temperature of ~40 °C—undergoes dramatic changes within this temperature range.

Based on the measured principal axis offset, we can gain insights into the broadband PER of the 469 m *t*DNANF coil with an $R_b$ of 6 cm within the range of 1525–1565 nm (see "Methods" and Supplementary Materials Section S2.3). As shown by the black squares in Fig. 3f, the estimated broadband PER remains at ~20 dB across the varying temperature range for the 10 m fibre segment. Additionally, the PER of the four-tube *t*DNANF coil was examined using a 40 nm bandwidth ASE source immediately after fibre winding. The resulting PER of 20.4 dB

(red star in Fig. 3f) confirms the high quality of the quadrupolar winding, despite the presence of complex twists and microbending conditions within the gyro. Notably, in ref. 49, a PER greater than 30 dB for a 467 m PBG-HCF with no bending was maintained only over a narrow wavelength range of 1530.0–1533.7 nm.

## Gyro test

The setup of our four-tube *t*DNANF gyro is illustrated in Fig. 4a. The ASE light source, based on a 3.2-m erbium-doped fibre, has an output power of 8.5 mW. A 1550-nm filter reflector and a gain flat filter are connected to the two ends to provide double-pass amplification and spectrum shaping, respectively, resulting in a bandwidth of ~40 nm centred at 1544 nm, as shown in Inset 1. Additionally, a fibre-optic isolator ensures monodirectional operation. A polarization-maintaining fibre coupler directs the light emitted from the source to a multifunction integrated optics chip (MIOC), which is fabricated on a lithium niobate substrate. The MIOC comprises a polarizer with a very high extinction ratio (~70 dB), a Y junction, and two push-pull electro-optic phase modulators for dynamic biasing. The outgoing ports of the Y junction are connected to the 469-m four-tube *t*DNANF coil via fibre-to-chip direct coupling using microlens pairs to circumvent mode size mismatch (see "Methods" and Supplementary Materials Section S3.2). The light returning through the MIOC is directed by the same fibre coupler to a photodetector (PD), whose output currents offer rotation rate information as well as feedback to control the square-wave biasing modulation–demodulation via an electronics package. The modulation depth is set at the maximum sensitivity point ($\pi/2$). The measured eigenfrequency of 320.47 kHz confirms that the effective refractive index of the four-tube *t*DNANF's fundamental mode is ~1.0, as expected since the fibre core mode travels mostly in air.

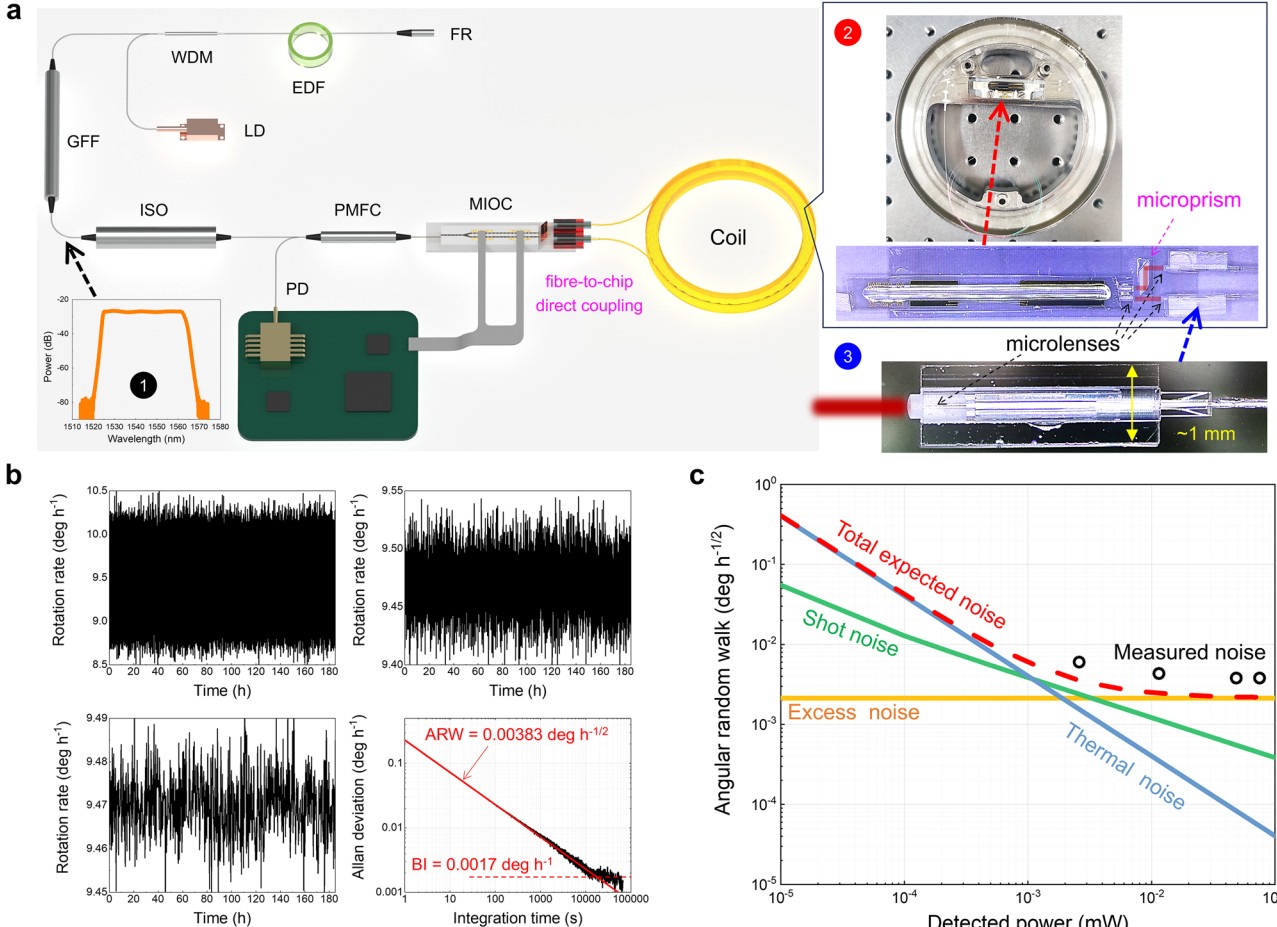

**Fig. 4 | Static performances of the four-tube truncated double nested anti-resonant nodeless fibre (*t*DNANF) gyro. a** Setup of the interferometric fiber optic gyroscope (IFOG). EDF erbium-doped fibre, LD 980 nm laser diode, WDM 980/1550 nm wavelength-division multiplexer, FR 1550 nm filter reflector, GFF gain flattening filter, ISO fibre-optic isolator, PMFC polarization-maintaining fibre coupler, MIOC multifunction integrated-optics chip, PD photodetector. Inset 1: Output spectrum of the amplified spontaneous emission (ASE) source. Inset 2: Photographs of the fibre coil and the MIOC with the two outgoing ports free-space coupled to the fibre coil. Inset 3: Photograph of a fibre end assembled with a microlens. **b** Static rotation rate readout from the gyro, displaying results over average time windows of 1 s, 100 s, and 1000 s, along with the Allan deviation of the data. BI bias instability, ARW angle random walk. **c** Measured ARW noise versus detected power for the gyro, compared with theoretically calculated noise of individual sources.

A microscopic magnification of the fibre-to-chip coupling assembly is shown in Inset 2 of Fig. 4a. It consists of two sets of microlenses and one reflective microprism. The microlens pairs function as beam collimators and mode field diameter adapters. The reflective microprism, placed between the two microlenses, ensures that the collimated beam from one outgoing port of the MIOC is reflected twice and is set far from the parallel beam emitted from the other port. This separation serves two purposes: (1) preventing crosstalk between the two light beams from the MIOC and (2) providing ample space for handling the two pigtails of the fibre coil, whose transverse sizes amount to ~1 mm after being assembled with microlenses (as shown in Inset 3 of Fig. 4a). The imbalance loss between the two pigtails (~0.2 dB) caused by the reflective microprism has a negligible impact on the gyro drift due to the extremely low Kerr nonlinearity in air-core fibres. The output facets of the MIOC and microlenses are 10°- and 8°-cleaved, respectively, to eliminate back-reflection, while the four-tube *t*DNANF is flat-cleaved. During fibre-to-chip assembly, we rotated one end of the fibre coil to maximize the polarization extinction ratio at the other end and then rotated the other end of the fibre coil to maximize the power at the PD, ensuring that the linearly polarized light exiting from the MIOC aligned with the principal axis of the four-tube *t*DNANF. Subsequently, without modulation-demodulation, the round-trip loss from the ASE source to the PD was measured to be 22.4 dB, and the power at the PD was 48.6 μW.

To assess the operation of the gyro, our four-tube *t*DNANF gyro was mounted on a stable pier with the coil axis perpendicular to the horizontal plane (see Supplementary Materials Sections S3.3 and S3.4). The 185-hour response of the gyro to the Earth's rotation at room temperature (25 °C) is displayed in Fig. 4b, which shows the results over average time windows of 1 s, 100 s, and 1000 s, accompanied by the Allan deviation of the data. The measured rotation rate agrees well with the theoretical value of ~9.47 deg h⁻¹, which is a fraction of the Earth's rotation rate, given that the latitude of the test site is 39°10′.

An ARW of 0.00383 deg h⁻¹/² and a BI drift of 0.0017 deg h⁻¹ can be determined from the Allan deviation curve[50], demonstrating that the resolution of our four-tube *t*DNANF gyro reached navigation grade. To provide a fair comparison, we adopt a commercial navigation-grade gyro, such as the solid-core PMF gyro in Fig. 2.22 of ref. 1, and normalize the ARW by the sensing coil area. Our ARW performance is very close to—less than twice worse than—that of classical IFOGs, which is excellent. The BI of the gyro is 29.4 times better than that reported in previous NANF-based RFOG[34]. The outstanding bias drift performance can be primarily attributed to the single-mode, low-loss light transmission in the coil (overall loss ~4 dB) and the wide spectrum band (FWHM ~40 nm) of the light source. The polarization-sustaining capability of our four-tube *t*DNANF coil, with a measured PER of 20.4 dB, as described in Fig. 3f, also contributes to the low gyro drift.

The noise in our four-tube *t*DNANF gyro primarily arises from three sources. As depicted in Fig. 4c, these contributions include excess noise from the ASE source (with an ARW proportional to $P^0$), thermal noise from the PD ($P^{-1}$), and shot noise ($P^{-1/2}$). Each noise component was calculated based on their respective specifications. The expected total noise (the red dotted curve in Fig. 4c) was determined by quadratically summing these statistically independent noise sources. Thermal noise is anticipated to dominate at detected powers below 1 μW, while excess noise becomes predominant above 3 μW. The total noise was experimentally validated by varying the output power of the ASE source. These ARW measurements (black circles in Fig. 4c) show very close agreement with the expected total noise. These ARW measurements (black circles in Fig. 4c) closely agree with the expected total noise, with only a slight discrepancy attributed to the wavelength dependence of the principal polarization axis of the four-tube *t*DNANF. This wavelength dependence induces spectral ripples in the gyro, which utilizes an all-polarization-maintaining waveguide configuration.

### Thermal sensitivity

Combining the *pure* Shupe[8] and Mohr[9] effects, a temperature- or strain-induced phase shift occurs between the two counter-propagating beams in the fibre coil of an IFOG. This phase shift is indistinguishable from the rotation-induced phase shift. The resulting rotation rate error ($\Omega_E$) can be expressed as[1]

$$\Omega_E = \frac{c}{LD} \int_0^{L/2} \{\langle n_{\text{eff}} S \rangle \cdot \Delta \dot{T} + \Delta[n_{\text{eff}} S] \cdot \langle \dot{T} \rangle\} \frac{L - 2z}{c/n_g} dz \qquad (1)$$

where $L$ and $D$ are the total length and diameter of the coil, respectively, and $n_{\text{eff}}$ ($n_g$), $S$, and $\dot{T}$ denote the effective (group) modal index, the Shupe constant, and the time derivative of temperature in a fibre segment of length $dz$ at a distance $z$ from one end of the coil. The symbol $\Delta$ stands for the difference between the two values at $z$ and $L - 2z$, while the angle brackets $\langle \rangle$ denote the mean of the values at these two sites.

The four-tube *t*DNANF gyro was stably placed on a vibration isolation base inside a temperature chamber to ascertain its response to transient temperature variation (see Supplementary Materials Section S3.5). The left panel of Fig. 5a shows the temperature of the chamber, the time derivative of temperature, and the rotation rate shift of the gyro under a nominal temperature change rate of 1 °C per minute within a temperature range of −40 to 60 °C. The temperature was tracked by a thermocouple attached to the supporting structure of the coil. The initial surface temperature of the coil was set at 20 °C, then lowered to −40 °C at the specified rate and maintained for 2 h before rising to 60 °C at the same rate and holding for another 2 h. During cooling, the coil temperature decreased steadily with a nonlinear ramping rate, depending on the temperature control accuracy of the chamber. The thermally induced rotation rate shift increased at the beginning of the temperature change and then decreased at the end. During heating, the rotation rate shift had a similar trajectory but with the opposite trend. This characteristic is dictated by both the heat conduction inside the fibre coil and the compensation mechanism to the thermally induced error by symmetrical quadrupole winding. The heat flow is gradually transferred from the outermost to the internal layers, yielding a series of thermally induced phase shifts. Despite partial cancellation of these shifts by means of quadrupolar winding, the overall phase shift continues to accumulate as the outer layers heat faster than the inner layers, resulting in a shift in the rotation rate, which gradually vanishes at the end of the temperature change. Theoretically, the thermally induced rotation rate shift is proportional to the temperature–time derivative. The measured rotation rate shift aligns well with this, verifying the gyro's thermal response.

The evolutions of the four-tube *t*DNANF gyro rotation rate shift at various temperature change rates of 0.2, 0.5, 1, 2, and 5 °C per minute are shown in the right panel of Fig. 5a. The maximum rotation rate shifts were 0.0353 deg h⁻¹ (at 5 °C per minute), 0.0162 deg h⁻¹ (at 2 °C per minute), 0.0079 deg h⁻¹ (at 1 °C per minute), 0.0051 deg h⁻¹ (at 0.5 °C per minute), and 0.0013 deg h⁻¹ (at 0.2 °C per minute), respectively, confirming that the maximum rotation rate shift increases with the temperature change rate. Similar experiments were conducted within the temperature ranges of 20–60 °C and −40 to 20 °C, revealing similar results (Fig. 5b, c).

To evaluate the thermal stability of the four-tube *t*DNANF gyro, a traditional PMF was quadrupolarly coiled with the same length and diameter to replace the four-tube *t*DNANF coil. The rotation rate shift evolution of the PMF gyro was similar to that of the four-tube *t*DNANF gyro during heating and cooling, but the PMF gyro was much more sensitive to transient temperature changes. Figure 5d shows the PMF gyro's maximum rotation rate shift under the same conditions as Fig. 5a. The solid-core PMF gyro exhibited a larger rotation rate shift than the air-core four-tube *t*DNANF gyro. For example, under a temperature change rate of 1 °C per minute, the maximum rotation rate shifts of the PMF and four-tube *t*DNANF gyros were 0.071 deg h⁻¹ and 0.0079 deg h⁻¹, respectively.

The thermal sensitivities of both gyros were determined by plotting the maximum rotation rate shifts as a function of the applied temperature change rate (Fig. 5e). The shifts varied linearly with the temperature change rate. Linear fitting revealed that the thermal sensitivity of the PMF gyro was approximately $0.0641/60 = 1.07 \times 10^{-3}$ deg °C⁻¹, while that of the four-tube *t*DNANF gyro was $0.0069/60 = 1.15 \times 10^{-4}$ deg °C⁻¹, $0.0060/60 = 1.00 \times 10^{-4}$ deg °C⁻¹, and $0.0094/60 = 1.57 \times 10^{-4}$ deg °C⁻¹ for temperatures ranging from −40 to 60 °C, 20–60 °C, and −40 to 20 °C, respectively. This indicates that the four-tube *t*DNANF gyro is 9.24 (−40 to 60 °C), 10.68 (20–60 °C) and 6.82 (−40 to 20 °C) times less sensitive to temperature variation over different temperature ranges than the PMF gyro under the same coil size and thermal shock conditions.

The wide-temperature scale factor measurement of the four-tube *t*DNANF gyro is carried out (see Supplementary Materials Section S3.6). The gyro exhibits excellent scale factor characteristics across a wide-temperature range (−40 to 60 °C), with a scale factor repeatability of 11.1 ppm, a maximum scale factor nonlinearity of 3.2 ppm, and a maximum scale factor asymmetry of 6.1 ppm. Therefore, a change in the gyro scale factor on the order of 10 ppm will not affect the thermal sensitivity experiment. The difference in the measured thermal sensitivities suggests that the thermal attributes of the materials used in the fibre coating and coil glue play a critical role. The key parameters of these materials, such as their viscoelastic properties and Young's modulus, change markedly with temperature.

In addition to the thermal excursion test, static measurements of the rotation rate for the four-tube *t*DNANF gyro at different temperatures and their Allan deviations are also presented in Fig. 5f. The long-term bias instabilities were 0.0035 deg h⁻¹ at 60 °C, 0.0017 deg h⁻¹ at 25 °C, and 0.0028 deg h⁻¹ at −40 °C. The slight discrepancy may arise from the thermally induced variations in the properties of the fibre coating or curing glue.

Table 1 compares the performances of representative air-core FOGs. The proposed four-tube *t*DNANF gyro outperforms all other air-core FOGs in terms of the ARW at high power, BS drift over 100 s, BI drift over a long enough period, and thermal sensitivity reduction compared to a PMF gyro of the same coil size. Our work demonstrated a dramatic 29.4-fold decrease in BI drift compared with any previous report. To the best of our knowledge, this is the first case of navigation-grade air-core FOG.

## Discussion

The use of an air-core fibre in FOG is expected to significantly reduce the extraneous phase drift and noise associated with the Kerr effect,

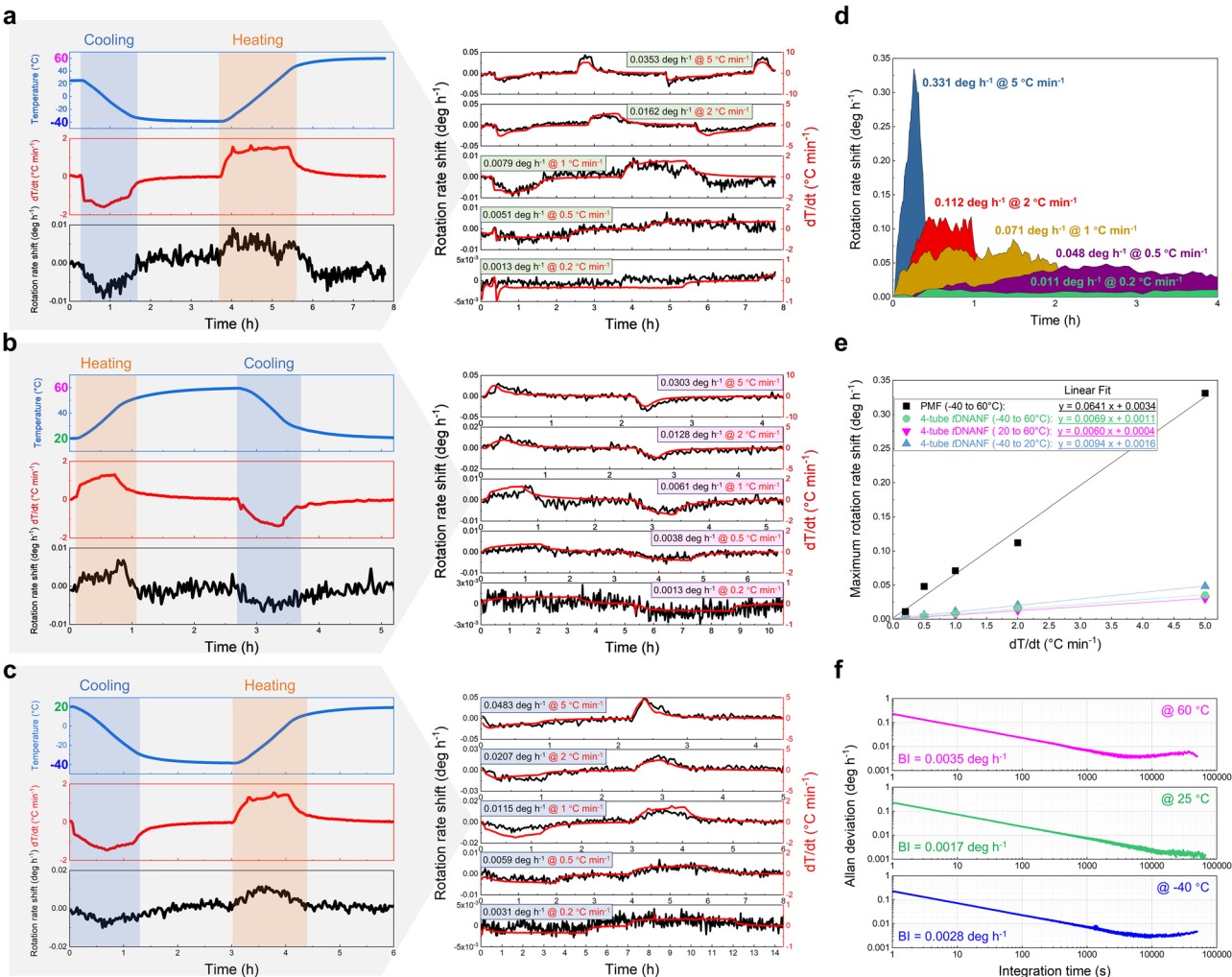

**Fig. 5 | Thermal stability of the four-tube truncated double nested antiresonant nodeless fibre (tDNANF) gyro compared to a conventional polarization-maintaining fibre optic gyroscope. a** Surface temperature of the four-tube tDNANF coil, temperature time derivatives (d$T$/d$t$), and rotation rate shifts of the four-tube tDNANF gyro under various temperature change rates of 0.2, 0.5, 1, 2, and 5 °C per minute within a temperature range of −40 to 60 °C. **b, c** Similar to **a**, but for the temperature intervals of 20–60 °C and −40 to 20 °C, respectively. **d** Temperature variation measurements for a gyro equipped with a conventional quadrupolar-wound polarization-maintaining fibre (PMF) coil (of the same length) subjected to identical conditions as in (**a**). **e** Correlation of the maximum rotation rate shift with the time derivative of temperature for both the conventional PMF gyro and the four-tube tDNANF gyro over different temperature ranges. **f** Allan deviations of the static rotation rate measured by the four-tube tDNANF gyro at −40 °C, 25 °C, and 60 °C. The centre panel replicates the data presented in Fig. 4b. BI bias instability.

Faraday effect, thermal effects, and radiation damage in sensing fibres[13,17]. These improvements could lead to higher long-term stability, lower noise, and better cost, size, weight, and power (C-SWaP) performance. However, early efforts faced challenges such as severe backscattering and poor spatial/polarization mode purity in PBG-HCFs. Although the introduction of shunt structures in cladding to achieve high spatial mode purity in PBG-HCFs was proposed a decade ago[49], the stringent requirements on fibre bending and orientation due to the small index mismatch between higher-order core modes and shunt cladding modes complicate the use of such fibres in a quadrupolar-wound IFOG.

Conversely, the strategy of degenerating polarization states of the core fundamental modes by varying the glass membrane thickness in the core surround with a fourfold rotational symmetry shape has proven effective in generic HCFs[43–45,49] and has been adopted in this work. Another advantage of this core surround design is that it aligns the principal axes of various wavelengths in the same direction, which is necessary for IFOG applications.

The inherent drawback of PBG-HCFs is that a great amount of modal field overlaps with the air/glass interfaces[51], which causes severe backscattering and interpolarization crosstalk, fundamentally undermining the Sagnac interferometric effect in an FOG. The long-standing stagnation of the low-loss PBG-HCF development can also be attributed to this issue[52].

The advent of ARF, which features broader bandwidth, better higher-order-mode suppression, and a greater laser damage threshold than PBG-HCF, has spurred numerous new application scenarios of HCF technology in communications[53], quantum state transmission[54], and high-power laser delivery[55]. In the field of FOG, decades of technique refinements have converged on the minimum-reciprocal-scheme IFOG, both in electronic and photonic aspects. Therefore, adhering to this well-established scheme and simply extending it, for example, through direct fibre-to-chip coupling with no redundant optics, may better advance the ultimate performance and promote widespread technology adoption. Recent attempts at ARF-based RFOG[33,34] cannot leverage these advantages.

**Table 1 | Representative air-core fibre optic gyroscopes (FOGs) and criteria of navigation-grade FOGs**

| Type of fibre and gyro system | Coil length/diameter (m) | ARW at high power[c] (deg h$^{-1/2}$) | Bias stability (deg h$^{-1}$) | Integration time (s) | Thermal stability | Ref. |
|---|---|---|---|---|---|---|
| 7-cell PBG-IFOG | 235/0.082 | – | 2 (drift) | / | 6.5 | 16 |
| 7-cell PBG-RFOG | 50 (Finesse = 7) | 0.96 | 20 | 10 | – | 57 |
| 7-cell PMPBG-IFOG | 250/0.05 | 0.048 | 0.51 | 50 | – | 58 |
| Kagome-RFOG | 5.6 (Finesse = 58.2)/0.13 | 0.04 | 0.15 | 200 | – | 33 |
| 6-tube NANF-RFOG | 136 (Finesse = 14.5)/0.114 | 0.09 | 0.05 | 3600-36000 | – | 34 |
| | | | 0.3 | 100 | | |
| 4-tube tDNANF-IFOG | 469/0.12 | 0.00383[a] | 0.0017 @ 25 °C | 20000 | 9.24/10.68/6.82[b] | This work |
| | | | 0.023[a] | 100 | | |
| Navigation-grade criteria | – | 0.017–0.0017 | 0.1-0.01 | 100 | – | |

[a]The values measured at −40 °C, 25 °C, and 60 °C are identical (see Fig. 5f).

[b]The ratios of thermal sensitivities between the solid-core PMF gyro and the air-core four-tube tDNANF gyro vary across the temperature ranges of −40 to 60 °C, 20–60 °C, and −40 to 20 °C (see Fig. 5e).

[c]A common approach to evaluating the angular random walk (ARW) performance of an interferometric fibre optic gyroscope (IFOG) involves normalizing it by the sensing coil area, specifically by considering the product of the ARW and the coil area. However, this method is not applicable to resonant fibre optic gyroscopes (RFOGs).

In summary, the 469 m four-tube tDNANF coil-based IFOG demonstrated in this work represents a significant step in the pursuit of high-performance air-core FOGs. With an ARW of 0.00383 deg h$^{-1/2}$ and a BI drift of 0.0017 deg h$^{-1}$, this gyro showcases navigation-grade performance above one order of magnitude better than the previous record[34]. Notably, a -10-fold reduction in thermal sensitivity, compared to that of a conventional PMF gyro with a coil of the same length, has been experimentally observed, thanks to the optimization of both the *pure* Shupe and Mohr effects, thereby updating previous predictions on the thermal stability gain of air-core FOGs[17–19]. The thermal stability is expected to further improve by optimizing the thermal and mechanical properties of the filling glue in the fibre coils[56].

Based on our analysis of fibre optical properties, the next goal is to further refine the phase birefringence and principal axis offset, which appear feasible within the context of our ARF design, potentially making air-core IFOGs a real contender to their solid-core fibre counterparts.

## Methods

### Finite element method simulation

The effective index, vector field distribution, and confinement loss of a polarization mode were simulated using commercial finite-element solvers, such as COMSOL Multiphysics, with an optimized mesh size and a perfectly matched layer. The geometrical parameters were extracted from SEM images of the fibre, with some adjustments made within the range of uncertainties. Dielectrics were modelled using the Sellmeier equation for silica and considering $n = 1$ for air. To assess the macrobend loss of a fibre, the refractive index distribution was conformally mapped to $n_b = n_s \cdot e^{x/Rb}$, where $n_b$ is the refractive index distribution of the bent fibre, $n_s$ is that of the straight fibre, $R_b$ is the bend radius, and the fibre is bent toward the negative x-axis.

### Fibre design and fabrication

To date, all reported fabricated NANFs and DNANFs adopt five or more tubular units, where each outer tube encloses a mid-sized tube and a smaller tube. These tubular units are radially oriented around a central core without touching each other, resulting in several void regions behind the gaps. Four-tube DNANFs are seldom considered because the corresponding four void regions are equivalent in size to the core area, promoting unwanted coupling of the fundamental core mode with the cladding air mode. To circumvent this effect, the four outer tubes are truncated into oversized semicircle shapes, pronouncedly diminishing the area of the voids behind the gaps. The fabrication of these advanced tDNANFs entails precision cutting of the full tubes at a prescribed angle by high-power lasers. Subsequently, these truncated tubes are aligned and assembled with the inner middle and smaller tubes, resulting in a structured assembly. The entire arrangement undergoes thermal drawing to form a preform, which is further processed into the final fibre. During the fibre drawing process, differential pressurization is strategically applied to four zones—the core region, the truncated outer tubes, the middle-sized tubes, and the smallest tubes—to ensure the integrity and performance of the fibre.

### Loss measurement

Spectral attenuation was measured by the cut-back method from 510 to 10 m for the four-tube tDNANF. A supercontinuum (SC-5, YSL, 470–2400 nm) source was stably butt-coupled to the fibre (inside a fibre splicer), which was looped on a bobbin with the circumstance of 1 m. The output of the fibre was connected to an optical spectral analyser (AQ6370D, Yokogawa, 600–1700 nm) through a magnetic clamp bare fibre adaptor. Multiple cleavages of the fibre end revealed little variation in the recorded spectra.

Macrobend loss measurements were conducted by comparing the transmission spectrum of a bent fibre with that of a quasistraight fibre with a loop radius of 50 cm.

### Symmetrical quadrupolar fibre winding

The four-tube tDNANF was wound from the centre, alternating layers from each half-length to position symmetrical segments in proximity. The quadrupolar winding method, where the fibre layer order is reversed pair by pair, can greatly circumvent time-transience related nonreciprocal effects of the entire coil. To achieve optimal results, the number of fibre layers should be a multiple of four, with each layer containing the same number of turns. During fibre winding, on-line transmission loss was monitored to adjust the tension in real-time, thus minimizing damage to the fibre.

### Spatial mode purity measurement

The modal purity of a fibre was analysed using the $S^2$ imaging method[48]. A tunable laser (Santec TSL-550A, 1490–1640 nm) transported light into the fibre under test, and an infrared CCD (WiDy Sens 320, NIT) camera determined the spatial distribution of the transmitted light as the wavelength swept across a range of 2.5 nm with a spacing of 1 pm. Subsequently, the multipath interference (MPI) at the fibre output, defined as the power of a higher-order mode relative to the fundamental mode, was calculated by applying Fourier transformation and then integrating the optical spectrum at each pixel in the cross section.

## Shupe constant measurement

The strain-free Shupe constant was measured using an all-fibre Mach–Zehnder interferometer (see Supplementary Materials Section S1.2). A 400 kHz linewidth tunable laser (Santec TSL-550A, @1550 nm) was employed. In the signal arm of the interferometer, the fibre under test (FUT) was coiled with a diameter of 6 cm and placed in a homemade thermal chamber along with two thermometers (with a resolution of 0.1 °C). Both ends of the FUT were spliced with SSMF pigtails and incorporated into the interferometer. The length difference between the SSMFs in the signal and the reference arms was less than 2 cm to minimize unwanted phase drift caused by SSMFs. After a $3 \times 3$ fibre optic coupler, the interferograms were acquired in real time by three InGaAs photodetectors (Thorlabs, PDA015C2) connected to a digital data acquisition card and then used to derive the optical phase change accumulated when light passes through the FUT.

## Polarization measurements

The polarization properties of our ARFs were measured by a crossed-polarizer setup, which consisted of a supercontinuum source, two calcite polarizers, two achromatic half-wave plates (HWP1/HWP2), and an optical spectrum analyser (see Supplementary Materials Section S1.3). The fibre under test was coiled and placed in a thermal chamber. The input and the output polarization states were tuned by rotating HWP1 and HWP2, respectively. Since the interpolarization coupling in ARFs is weak, a coiled ARF can be regarded as a wave plate with a wavelength-dependent principal axis angle ($\theta_A$) and phase retardance ($\phi$) (see Supplementary Materials Section S2.1). Over a modestly wide wavelength range (e.g., 1525–1565 nm in this study), the intensities transmitted through a crossed-polarizer setup can be approximately expressed as[38]

$$\begin{cases} I_{0°/0°}(\lambda) = 1 - \frac{1}{2}\sin^2(2\theta_A(\lambda)) \cdot (1 - \Delta \cdot \cos(\phi(\lambda))) \\ I_{0°/90°}(\lambda) = \frac{1}{2}\sin^2(2\theta_A(\lambda)) \cdot (1 - \Delta \cdot \cos(\phi(\lambda))) \\ I_{45°/45°}(\lambda) = 1 - \frac{1}{2}\cos^2(2\theta_A(\lambda)) \cdot (1 - \Delta \cdot \cos(\phi(\lambda))) \\ I_{45°/135°}(\lambda) = \frac{1}{2}\cos^2(2\theta_A(\lambda)) \cdot (1 - \Delta \cdot \cos(\phi(\lambda))) \end{cases} \quad (2)$$

Here, the subscripts 0°/0°, 0°/90°, 45°/45°, and 45°/135° represent the input/output polarization angles of measurement, and an empirical less-than-one factor $\Delta$ accounts for the lossy interpolarization coupling between sequential segments along the fibre. From the measured spectra, the principal axis angle ($\theta_A$) can be derived from Eq. (2) using the same method as in ref. 38 (the results of which are presented in Fig. 3e). Based on the measured $\theta_A(\lambda)$, a broadband PER (integrated from $\lambda_1$ to $\lambda_2$) can be assessed by the following expression when the phase retardance $\phi$ varies rapidly with wavelength considering a fibre length of hundreds of metres (the results of which are presented in Fig. 3f):

$$PER(dB) \approx 10 \times \log \frac{\int_{\lambda_1}^{\lambda_2}[1 - \frac{1}{2}\sin^2(2\theta_A(\lambda))]d\lambda}{\int_{\lambda_1}^{\lambda_2}[\frac{1}{2}\sin^2(2\theta_A(\lambda))]d\lambda}. \quad (3)$$

## Fibre-to-chip direct coupling

Direct coupling assembly is used to prevent microstructure collapse of the four-tube $t$DNANF due to fusion. Two sets of microlenses are introduced between the pigtails of the four-tube $t$DNANF coil and the bare Y waveguide chip to collimate and converge the light, facilitating low-loss coupling. Additionally, a reflective microprism is placed between one set of microlenses. The collimated beam emitted from the microlenses is reflected twice, ensuring that the parallel light beams transmitted through the two pairs of microlenses are spatially separated. This separation serves two purposes: (1) it increases the transverse distance to prevent crosstalk between the two light beams from the Y waveguide, and (2) it provides sufficient space to facilitate direct coupling operations of the four-tube $t$DNANF.

## Gyro static performance measurements

To measure the FOG static performance, the four-tube $t$DNANF gyro was mounted on a stable pier at room temperature (25 °C) with an input rotation rate of only a fraction of the Earth's rotation rate, approximately 9.47 deg h⁻¹. At such a low and stable input rate, we can ascribe any drift or variation to random noise (ARW) over short integration times or bias stability over longer integration times. The random measurement signal of the gyro, $\Omega(t)$, can be averaged over an integration time $\tau$, yielding a series of averaged values $\overline{\Omega_k}(\tau)$. The Allan deviation, ADEV($\tau$), is defined as the square root of half the mean value of the square of the difference between two successive averaged values:

$$\text{ADEV}(\tau) = \sqrt{\frac{1}{2}\langle(\overline{\Omega_{k+1}}(\tau) - \overline{\Omega_k}(\tau))^2\rangle} \quad (4)$$

The Allan deviation indicates that the rate output variation of the gyro (rate uncertainty) is a function of the integration time. The ARW can be obtained from the Allan deviation value at an integration time of 1 s divided by 60. The bias stability at an arbitrary integration time is equal to the Allan deviation at the corresponding integration time. Bias instability appears on the Allan deviation plot as a flat region around the minimum.

## Thermal excursion measurements

The gyro was stably placed on a vibration isolation base inside a temperature chamber (GWS, WKZT1-50C). The thermally induced rotation rate shift (thermal stability) was measured at various temperature change rates of 0.2, 0.5, 1, 2, and 5 °C per minute across different temperature ranges: −40 to 60 °C, 20–60 °C, and −40 to 20 °C.

## Data availability

The experimental and numerical data generated in this study have been deposited in the Figshare.

## Code availability

The code used in this paper to process data can be accessed in an open data-storage platform [https://doi.org/10.6084/figshare.28050587].

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

## Acknowledgements

This work was supported by the National Natural Science Foundation of China (No. 62075083 to W. D., No. 12074349 to M. L., Nos. 62222506 and U21A20506 to Y. W., No. 62105122 to S. G.), the Basic and Applied Basic Research Foundation of Guangdong Province (No. 2021B1515020030 to Y. W., No. 2021A1515011646 to S. G., No. 2022A1515110218 to Y. S.), the Guangzhou Science and Technology Program (No. 202201010460 to Y. S.), and the Major Key Project of Pengcheng Laboratory (W. D.).

## Author contributions

W.D. and M.L. conceived the study. W.D. and X.Z. supervised the project. Y.S., Q.H., and H.W. designed and conducted the fibre optical characterization with assistance from W.D. S.G. and Y.W. designed and fabricated the *t*DNANF and NANF. H.C. and Y.S. performed numerical simulations with support from W.D. and Y.W. M.L., F.H., M.Y., W.L., and X.Z. carried out the quadrupolar fibre winding and gyro test. M.L. and F.H. executed the fibre-to-chip direct coupling and temperature sensitivity measurements. W.D., M.L., and Y.S. prepared the figures and drafted the manuscript. M.L., W.D., and F.H. conducted additional experiments, made revisions, and responded to reviewers' comments.

## Competing interests

The authors declare no competing interests.
