## [Transparent Peer Review file · Nature Communications]

Please wait...

If this message is not eventually replaced by the proper contents of the document, your PDF viewer may not be able to display this type of document.

You can upgrade to the latest version of Adobe Reader for Windows®, Mac, or Linux® by visiting http://www.adobe.com/go/reader_download.

For more assistance with Adobe Reader visit <http://www.adobe.com/go/acrreader>.

Windows is either a registered trademark or a trademark of Microsoft Corporation in the United States and/or other countries. Mac is a trademark of Apple Inc., registered in the United States and other countries. Linux is the registered trademark of Linus Torvalds in the U.S. and other countries.

Re: ID: NCOMMS-24-41217A-Z

Title: Navigation-grade interferometric air-core antiresonant fibre optic gyroscope with enhanced thermal stability

Authors: Maochun Li^{1,†}, Yizhi Sun^{2,3,5†}, Shoufei Gao^{2,3,5†}, Xiaoming Zhao^{1,*}, Fei Hui¹, Wei Luo¹, Qingbo Hu^{2,3}, Hao Chen^{2,3}, Helin Wu^{2,3}, Yingying Wang^{2,3,5}, Miao Yan^{1,4}, and Wei Ding^{2,3,5,6*}

Response to reviewer 1

General Comments: *In this work, Li et al. demonstrate an interferometric fiber-optic gyroscope (IFOG) using a truncated dual nested hollow core antiresonant nodeless fiber (tDNANF). They highlight the many advantages of hollow core fibers for improving performance in fiber gyroscopes, and they show an impressive collection of fiber and gyro level performance results, though several of these results are poorly evidenced. The work is essentially divided into two sections – the first describing the fiber properties and the second describing the gyro properties.*

In the fiber/coil property section, there are several demonstrations:

· *Their 4-fold microstructure is shown to have less wavelength dependence in the birefringence axes of some comparable 5 and 6 fold structures.*

· *Polarization extinction ratio (PER) in their 4-tDNANF coil is shown to have a value of 20 dB via a combination of measurement and analysis. This PER represents a value akin to gyro coils on the market.*

· *The fiber bend loss is measured several ways and the total loss of the coil from these measurements is estimated to be on the order of ~3-5 dB/km in the 6cm radius of the coil. This is comparable to the current state of the art in hollow core fibers (HCFs).*

· *Their 469 m long coil shows no evidence of higher order mode propagation to within the -70 dB limit of their measurement – this is akin to other higher order mode suppression results reported on antiresonant HCFs in the literature.*

· *They measure a significant reduction (14.4X) in the thermal phase sensitivity of their 4-tDNANF over a solid core PM fiber. This is akin to other comparisons in the literature for antiresonant HCFs.*

In the gyro performance section, the authors give the two most important results of this work:

· *They show drastic improvement in rate bias thermal instability – a key environmental performance limiter in FOGs. Their results are over wide temperature ranges and show the HCF IFOG outperforming a comparable solid core IFOG by 6-10X*

· *They demonstrate gyro rate bias instability – one of the key metrics of gyro performance – which surpasses previous results by at least an order of magnitude for the case of hollow core fiber gyro implementations (a comparison with solid fiber results is not given).*

These results are a significant addition to the field of fiber optic gyroscopes, in which hollow core fibers offer substantial promise to improve performance and reduce environmental effects. The gyro level results are impressive and represent significant progress for a fiber sensor that is extremely important to the future of guidance and navigation. The IFOG is among the most exacting use cases for an optical fiber, and thus these results can be interpreted as a significant achievement in the wider context of optical fiber applications.

Response. We thank the reviewer's positive general evaluation to our work that "...they show an impressive collection of fiber and gyro level performance results... These results are a significant addition to the field of fiber optic gyroscopes... The gyro level results are impressive and represent significant progress... these results can be interpreted as a significant achievement in the wider context of optical fiber applications" We will carefully revise the manuscript complying with the raised valuable critical feedback.

Regarding the reviewer's suggestion of "...a comparison with solid fiber results...", we would like to clarify that our control solid-core gyroscope, which offers equivalent precision, was selected from our company's batch of products. We specifically chose the unit with the best temperature characteristics as the comparison standard. As Reviewer 2 pointed out, winding errors may randomly compensate, resulting in exceptionally favorable temperature characteristics in some coils. However, this is a low-probability event, and generally, our coil manufacturing process remains stable within a consistent range.

Additionally, Reviewer 2 has helped us fairly compare the ARW performance of our air-core gyro with that of commercial navigation-grade IFOGs (see below).

Changes in the revised manuscript:

In the fourth paragraph of the Section "**Results**", Subsection "**Gyro test**", we have added "To provide a fair comparison, we adopt a commercial navigation-grade gyro, such as the solid-core PMF gyro in Fig. 2.22 of Ref. 1, and normalize the ARW by the sensing coil area. Our ARW performance is very close to---less than twice worse than---that of classical IFOGs, which is excellent".

Additionally, a new footnote ^c has been added to Table 1 to clarify the same meaning with "^c a common approach to evaluating the ARW performance of an IFOG involves normalizing it by the sensing coil area, specifically by considering the product of the ARW and the coil area. However, this method is not applicable to RFOGs".

However, in each of the key performance areas, there are major gaps in the evidence which is used to justify the claimed results. In my opinion, the manuscript cannot be published without very significant revision to the text to better substantiate

these claims. In particular, there are four areas which must be addressed to provide more substantial verification of the performance numbers quoted:

Comments 1. *One of the headline results is the author’s demonstration of 6.8 to 10.7 times reduction in thermal sensitivity of the gyro rate bias over an equivalent test using conventional PM solid core fiber. The authors measure the thermal phase variation of a 10m coil of their HCF and show that this is 14.4 times lower than in their PM fiber control sample. This data is used to support the apparent reduction in gyro level thermal rate bias variation. However, for a coil of the dimensions used in their gyro demonstration (469m, 36-layer, 6-cm radius), the Shupe Effect is not the limiting error mechanism in the thermal rate bias performance of the coil. The dominant phase error arises from layer-to-layer thermal pressure causing circumferential strain in the fiber coil, also known as “Mohr Effect” [1], [2]. This effect is not captured in the authors’ surrogate test, which employs only 10m of fiber. For the Mohr Effect, the use of a hollow core fiber having identical coating materials and coil winding, compared to a geometrically identical solid core PM fiber, would only benefit to the extent of the difference in effective index squared – i.e. a factor of ~2. The authors must provide justification which explains this drastic and unexplained reduction in Mohr Effect for their HCF coil. If this arises from a difference in mechanical or geometric properties between the HCF and PMF, then the 6.8-10.7X reduction claimed would not seem to be a fair comparison. If it arises from some other feature of the HCF or HCF coil structure, then that should be detailed in the text.*

Response. Thanks for the reviewer’s comments. As Reviewer 2 also pointed out, our previous draft missed the discussion of “Mohr Effect”, which should be the leading reason of the observed reduction in thermal sensitivity of our gyro. In the revised manuscript, a detailed explanation to both “pure Shupe Effect” and “Mohr Effect” is added to avoid misunderstanding.

Complying with the nomenclature in Ref. [1], the pure Shupe effect (*delta T dot* effect, $\Delta\dot{T}$) ignores the strain applied on fiber, while the Mohr effect (*T dot* effect, \dot{T}) emphasizes the differential-strain-induced Shupe constant (*S*) variation. It is worth to note that the equation (6.3) in Ref. [1] of $\alpha_T = \frac{dn}{dT} + n\alpha_{fiber}$ (thermal coefficient of the phase) is equivalent to nS . Here, the Shupe constant is defined as the *relative* phase change with temperature (see Ref. [19], 2007-JLT-paper), $S = 1/\varphi \cdot d\varphi/dT$, and n stands for effective modal index. So, α_T is in unit of [$^{\circ}\text{C}$], and S is in unit of [ppm/ $^{\circ}\text{C}$]. This discrepancy also explains the difference between the n dependence in Eq. (6.12) of Ref. [1], $\delta\phi_{Sh}(z) = \frac{2\pi}{\lambda} (\langle\alpha_T\rangle \cdot \Delta\dot{T}) \frac{L-2z}{c/n} \delta z$, and the n^2 dependence in Eq. (1) of Ref. [19], $\Delta\phi_E = \frac{2\pi}{\lambda_0 c} S n^2 \int_0^L (L - 2z) \dot{T}(z) dz$. However, we want to emphasize that one n in these two equations should stand for group index (n_g), instead of effective index (n_{eff}). It is

known that $n_g > n_{\text{eff}}$, and in a vacuum-core HCF $n_g > 1$ to be in accordance with the Theory of Relativity.

Since our thermal sensitivity characterizations to gyros (in Fig. 5e) collect the *maximum* rotation rate shifts across a wide range of time derivative of temperature (dT/dt), it is hard to judge that all these *maxima* occur in the second-order time derivative ($\ddot{T} \neq 0$) region (corresponding to a change of temperature excursion rate, where the pure Shupe effect dominates) or in the first-order time derivative ($\dot{T} \neq 0$) region (where the Mohr effect dominates), the revised manuscript addresses both effects and their influences to our gyro. Thanks Reviewer 2 for reminding that there is a very clear picture in Fig. 6.5 of Ref. [1] for this problem.

Changes in the revised manuscript:

In the second paragraph of the Section “**Introduction**”, the sentence has been revised to “**such as the temperature and strain induced time-transience effects^{8,9}**”.

After the fourth paragraph of the Section “**Results**”, Subsection “**Fibre optical characterization**”, a new paragraph has been added to address the issue of strain-induced variations in the Shupe constant within a coil, specifically the Mohr effect. We categorize the time-transient nonreciprocal effects into two temporal regions ($d^2T/dt^2 \neq 0$ and $dT/dt \neq 0$) and assess the magnitude of thermally induced strains by calculating fiber stiffness. Greater stiffness corresponds to higher resistance to the Mohr effect. The added paragraph is “**However, it is widely recognized that in a real fibre coil, the Shupe constant is not uniform due to differential longitudinal strains. When the coil is in the temperature regions with a non-zero first-order time derivative ($dT/dt \neq 0$), the Mohr effect---caused by layer-to-layer variation in the Shupe constant under changing temperature---becomes the primary source of thermal insensitivity^{1,9}. To evaluate the impact of the Mohr effect, we calculate the fibre stiffness, defined as the product of Young’s modulus and cross-sectional area. Our four-tube tDNANF exhibits a 7.07-fold increase in stiffness compared to conventional PMF (see Supplementary Material S1.4). Combined with the beneficial factor of $n_g n_{\text{eff}} \times$, the strain induced time-transient effect in our tDNANF gyro is expected to be ~ 14 times smaller than that of an SCF gyro**”.

Throughout the manuscript, we highlight that our fiber optical characterization provides “**the strain-free Shupe constant**”.

Furthermore, in this work, we adopt the experimental results of Ref. [32] for comparing our Shupe constant measurement. In that study, the authors accounted for the cross-sectional area considerations of state-of-the-art AR-HCFs. Our fiber geometry and measurement results closely align with their investigations. It is important to distinguish between the phase thermal coefficient (S_φ) per unit physical fiber length---i.e., α_T as defined in Eq. (6.3) of Ref. [1]---and the Shupe constant (S) for relative phase. The measurements from Ref. [32] and our study suggest that the value $dn/dT = 8.5 \times 10^{-6}/^\circ\text{C}$ in Ref. [1] for the index of silica (see Eq. (6.2)) warrants further verification.

Comments 2. *The authors devote significant attention to fiber and coil level measurements which are used to assert that the polarization extinction ratio (PER) of their coil is thermally stable and achieves a level of 20 dB. Here the evidence used to justify this 20 dB PER claim comes from a patchwork of surrogate measurements and analyses:*

- *A measurement on the full 469 m coil, using a 10 nm bandwidth light source shows PER of 20.5 dB.*

- *Modelling of the effective birefringence axis from a series of crossed polarizer measurements on a 10m fiber measured over temperature is used to compute an equivalent PER over temperature of ~20 dB.*

- *Measurements of 1.7m, 3m, and 10m lengths of fiber are used to assert that the birefringence axis variation remains low.*

20 dB PER is a reasonable benchmark for a gyro coil, but for commercial gyro coils measurements are always specified over the full length of the coil, and at least over the operating wavelength range of the gyro light source. The PER measurement which the authors present for the full gyro coil was done with 10nm bandwidth rather than the full 40nm bandwidth of the gyro light source. Reported PER results must cover the full range of the gyro light source, or else they do not serve as a useful benchmark. Likewise, the extrapolation of birefringence results from a measurement on 10m of fiber is unsuitable. The twist state of 1.7m, 3m, and 10m coils is extremely different from that of a 469 m long quadrupole-wound gyro having several thousand winds. Here the wavelength dependent birefringence vector will arise from a hugely complex overlap of effective retardances unless the orientation of the microstructure is precisely controlled relative to the coil winding axis. Extrapolation of a 10m measurement is misleading and should not be asserted to be representative of the full coil as it is in Fig. 3f. Use of these surrogate analyses and results convolutes the work unnecessarily, and it is a bizarre and illogical replacement for an extremely simple measurement – the surrogate results should be replaced with the actual PER measurement covering the full gyro operating spectrum and on the actual gyro coil. This should be accompanied with some discussion of how the apparent PER should contribute to polarization errors and hence rate bias instability in their HCF IFOG.

Response. Thank you for the comments. Upon re-examining our original experimental logs (see below), we discovered that the PER of the coil was measured using two different light sources with bandwidths of 10 nm and 40 nm. Following the reviewer's suggestion, it is more appropriate to report the PER value as 20.4 dB when using the 40 nm bandwidth ASE source.

Changes in the revised manuscript

We have updated the main text and the caption of Fig. 3 to reflect the revised ASE source bandwidth and the updated PER value.

We acknowledge the reviewer’s observation that the PER measurement for a standalone 10 m fiber differs from that of the full gyro coil. Please refer to the term “ultimate limit” in the sixth paragraph of the Section “**Results**”, Subsection “**Fibre optical characterization**”. In the latter part of that paragraph, we also discuss the influence of microbending conditions.

In the revised manuscript, we have added the phrases “for the 10 m fibre segment” and “despite the presence of complex twists and microbending conditions within the gyro” in the seventh paragraph to highlight the difference between the black squares and the red star in Fig. 3f.

Additionally, we have removed the phrase “closely agrees with the estimation” to correct the logical inconsistency.

Comments 3. Another headline result of this work is the gyro bias instability, which the authors identify as 0.0014 deg/hr. However, the results shown in the Allan deviation would indicate a significantly higher bias instability perhaps as much as 2-3X. The value of 0.0014 deg/hr appears to come from a near minimum value of the Allan deviation measured near 8500 seconds. This is an incorrect and unexplained representation of the bias instability. To justify this number, the authors should provide further test data demonstrating that 0.0014 deg/hr represents a real long-term expectation value for the gyro bias instability, or they should revise the quoted bias instability to be inline with the calculation from published standards such as the IEEE

Specification of IFOGs [3]. Likewise, the bias instabilities shown in Fig. 5f should be updated to be inline with a standard calculation, as these also appear to be underestimates. The angle random walk (ARW) is also derived from an overly simplistic method, and the standard calculation method should be used instead, though in this case the reported value is likely very close to that which will be derived from a standardized calculation.

Response. We thank the reviewer for highlighting concerns regarding the gyro bias instability measurement. To address this, we have conducted an extended 185-hour long-term bias test, replacing the initial 24-hour data. This extended testing period allows for a more accurate assessment of both the ARW and BI.

Using the Allan Variance method in accordance with IEEE Std 952-2020, we recalculated the ARW to be $0.00383 \text{ deg h}^{-1/2}$ and the BI drift to be $0.0017 \text{ deg h}^{-1}$. These revised values are consistent with our previously reported results but offer enhanced accuracy due to the longer testing duration. We have also updated the bias instability values shown in Fig. 5f to ensure they are in line with standardized calculations.

Changes in the revised manuscript

The Fig.4b has been revised to reflect the updated bias instability measurements derived from the 185-hour test. In the fourth paragraph of the Section “**Results**”, Subsection “**Gyro test**”, the sentence has been updated to “**An ARW of $0.00383 \text{ deg h}^{-1/2}$ and a BI drift of $0.0017 \text{ deg h}^{-1}$ can be determined from the Allan deviation curve⁵⁰**”. The corresponding ARW and BI data throughout the manuscript have been updated to incorporate these more accurate measurements. Fig. 5f has been updated.

Comments 4. *The authors do not mention any assessment of the gyro scale factor or rate bias offset in this work. Such a measurement is essential to give confidence that the measured phase is indeed a Sagnac phase rather than merely a large bias offset. Without such a measurement, the data can easily be misinterpreted or misread. Even a multi-rate measurement using different components of the Earth’s rotation rate would be sufficient to ensure that the signal measured here is indeed the Sagnac phase. At minimum, the measure rate bias offset and apparent phase scale factor should be reported.*

Response. We appreciate the reviewer’s insightful feedback regarding the assessment of the gyro scale factor and rate bias offset. To address this concern, we conducted comprehensive scale factor tests of the 4-tube tDNANF gyro across a wide temperature range ($-40 \text{ }^\circ\text{C}$ to $60 \text{ }^\circ\text{C}$) during our temperature sensitivity experiments. Although the scale factor analysis was not the primary focus of this study and thus was not initially reported, our findings demonstrate excellent scale factor characteristics: a scale factor repeatability of 11.1 ppm, a maximum scale factor nonlinearity of 3.2 ppm, and a maximum scale factor asymmetry of 6.1 ppm.

These results confirm that the measured phase is indeed the Sagnac phase rather than a large bias offset, ensuring the reliability and accuracy of our gyro measurements. To further enhance transparency and provide comprehensive details, we have included the scale factor test procedures and data in the Supplementary Material (Section S3.6). This additional information underscores the robustness of our gyro's performance and aligns with the reviewer's suggestions for ensuring the validity of our measurements.

Changes in the revised manuscript

In the sixth paragraph of the Section “**Results**”, Subsection “**Thermal sensitivity**”, we have added the following sentence to elaborate on the scale factor measurements “The wide-temperature scale factor measurement of the four-tube *t*DNANF gyro is carried out (see Supplementary Material S3.6). The gyro exhibits excellent scale factor characteristics across a wide-temperature range (-40 to 60 °C), with a scale factor repeatability of 11.1 ppm, a maximum scale factor nonlinearity of 3.2 ppm, and a maximum scale factor asymmetry of 6.1 ppm. Therefore, a change in the gyro scale factor on the order of 10 ppm will not affect the thermal sensitivity experiment.”.

In the **Supplementary Material**, we have added a new section of **S3.6** to detail the wide-temperature scale factor test process and results. Please see “Under a nominal temperature change rate of 1 °C per minute, spanning a temperature range from -40 °C to 60 °C, a total of seven scale factor tests were conducted on the four-tube *t*DNANF gyro. Each test was carried out over a 10-minute interval, with rotation rates set to $\pm 0.2^\circ/\text{s}$, $\pm 0.5^\circ/\text{s}$, $\pm 1^\circ/\text{s}$, $\pm 2^\circ/\text{s}$, $\pm 5^\circ/\text{s}$, $\pm 10^\circ/\text{s}$, $\pm 20^\circ/\text{s}$, $\pm 50^\circ/\text{s}$, $\pm 100^\circ/\text{s}$, $\pm 200^\circ/\text{s}$, $\pm 300^\circ/\text{s}$, and $\pm 400^\circ/\text{s}$, along with a 40 second dwell time for each rate. The rotation rates were systematically increased from lower to higher values, beginning with forward rotation followed by reverse rotation.

Figure S13 presents the results of these seven scale factor measurements across the wide temperature range for the four-tube *t*DNANF gyro. Specifically, Fig. S13a shows the gyro outputs corresponding to the various rotation rates, Fig. S13b illustrates the gyro temperature changes during the tests, and Fig. S13c displays the linear regression analysis of the seven measurement sets. The analysis reveals a scale factor repeatability of 11.1 ppm, a maximum scale factor nonlinearity of 3.2 ppm, and a maximum scale factor asymmetry of 6.1 ppm. These results confirm the gyro's excellent scale factor characteristics across the tested temperature range, ensuring that variations in the scale factor on the order of 10 ppm do not significantly impact the thermal sensitivity experiments.”

Fig. S12. Wide-Temperature Scale Factor Measurement. (a) Gyro outputs corresponding to different rotation rates. (b) Gyro temperature changes during measurements. (c) Rotation measurement results for the seven test sets. (*Note: A 100 % offset was artificially added between each set for clarity purposes.*)

Comments 5. *Along with these major concerns, there are several areas where I would recommend that the authors make modifications or elaborations to improve the text:*

- The authors state that the measured ARW “shows very close agreement with the expected total noise” but the noise given in Fig 4c is roughly 2 times that of the limiting contribution from light source excess noise. Can the authors account for this discrepancy?*

Response. The wavelength-dependent variation of the principal polarization axis direction in the four-tube tDNANF (as illustrated in Fig. 3e of the main text, the principal axis angle shifts by $\sim 10^\circ$ across the wavelength range of 1525–1556 nm) induces ripples in the light source spectrum after transmission through the MIOC and our 469-m four-tube tDNANF coil. This increases the parasitic interference. We conjecture that this effect is responsible for the measured gyroscope noise being roughly twice the expected value.

Several factors influence the extent of the observed light source ripples, including the wavelength-dependent shift of the principal polarization axis, the polarization rejection capabilities of the MIOC, and the distribution and magnitude of polarization mode coupling (crosstalk) within the four-tube tDNANF coil.

Panel (a) of the figure below shows the measured light source spectrum before PMFC, as shown in the inset 1 of Fig. 4a in the main text, and after PD (see Fig. 4a, after passing through the MIOC and the four-tube tDNANF coil), revealing the generation of

noticeable ripples. These data are secured from another worse four-tube tDNANF coil (of 530 m length) exhibiting pronounced property degradation—approximately 10-fold.

Panels (b) and (c) display the interference fringes derived from the spectra at PMFC and PD, respectively. It can be seen that numerous interference peaks appear on both sides of the main interference peak due to the rippled light source, leading to parasitic interference with the main peak and thus affecting the accuracy of the gyroscope.

(a) Light source spectrum before and after passing through the MIOC and the four-tube tDNANF coil (data from another four-tube tDNANF gyro exhibiting significant PER and drift degradation—approximately 10-fold). (b) Interference fringes without ripple. (c) Interference fringes with ripple.

Changes in the revised manuscript

We have revised the manuscript accordingly. Please refer to the last paragraph of the Subsection “Gyro test”. Please see “These ARW measurements (black circles in Fig. 4c) closely agree with the expected total noise, with only a slight discrepancy attributed to the wavelength dependence of the principal polarization axis of the four-tube tDNANF. This wavelength dependence induces spectral ripples in the gyro, which utilizes an all-polarization-maintaining waveguide configuration”.

· There are several measurements of bend loss, and the excess loss during coiling is reported, but the final, total loss of the coil is never given. Did the authors measure the coil loss? If so, I would suggest adding this measurement to the text to avoid confusion.

Response. Thanks for the reviewer’s comments. We have measured the total loss of the coil, which amounts to 4.23 dB. This total loss is comprised of 0.19 dB fiber background loss, 1.51 dB additional loss due to winding, and 2.53 dB additional loss resulting from potting adhesive curing.

Changes in the revised manuscript

We have updated the manuscript accordingly. Please refer to the second paragraph

of the Subsection “**Fibre optical characterization**” . Please see “An overall extra loss of 1.51 dB was detected during winding when the 469 m fibre length was wound with 36 layers and 30 turns, featuring an inner radius of 6 cm and an outer radius of 7.6 cm. The linear fit in Fig. 3b indicates an additional loss of 3.3 dB km^{-1} , which is consistent with the measured macrobend loss at $R_b = 6 \text{ cm}$. After curing, the coil yielded a total loss of 4.23 dB, comprising 2.53 dB additional loss from potting adhesive curing, 1.51 dB additional loss from winding, and 0.19 dB fibre background loss.”

· I believe the aim of Figure 1 is to clarify the context and show areas in which the use of hollow core fiber can improve various FOG architectures. However, the figure is confusing, convoluted, and does not add to understanding of the work. I would suggest the authors either find another strategy to convey this information effectively visually, or dispense with this figure entirely.

Response. We thank the reviewer for this thoughtful suggestion. To reduce the confusion and complexity present in the previous version of Figure 1, we have redesigned the figure's layout and presentation to enhance clarity. The updated figure more effectively highlights the contexts and areas where the use of hollow core fiber can improve various FOG metrics, thereby contributing to a better understanding of our work.

· The authors refer to the “minimum scheme” of the IFOG in several places – I believe they are intending to refer to the “minimum reciprocal scheme”.

Response. We have made revision. Please see the first paragraph of the Section “**Introduction**” and the fourth paragraph of the Section “**Discussion**”.

Comments 6. *I look forward to hearing the author’s responses in due course. The results which have been achieved here are important gyro performance demonstrations which substantially improve on the state-of-the-art. Assuming the addition of the further evidence and analytical rigor identified above does not materially change these performance results, then I believe these results could be published in Nature Communications.*

Response. We sincerely appreciate the reviewer’s thoughtful and careful evaluation of our work. Your valuable feedback has significantly enhanced the quality of this manuscript. We have carefully incorporated your suggestions in the revised version and are deeply grateful for your contributions. Thank you for recognizing the importance of our gyro performance demonstrations and for your support in considering our work for publication in *Nature Communications*.

References:

- [1] F. Mohr and F. Schadt, "Rigorous treatment of fiber-environmental interactions in fiber gyroscopes," in *2008 IEEE Region 8 International Conference on Computational Technologies in Electrical and Electronics Engineering*, Jul. 2008, pp. 372–375, doi: 10.1109/SIBIRCON.2008.4602634.
- [2] J. Pillon et al., "Thermomechanical analysis of the effects of homogeneous thermal field induced in the sensing coil of a fiber-optic gyroscope," *Finite Elem. Anal. Des.*, vol. 212, no. July, p. 103826, Dec. 2022, doi: 10.1016/j.finel.2022.103826.
- [3] *IEEE Std 952-2020 -- IEEE Standard for Specifying and Testing Single-Axis Interferometric Fiber Optic Gyros*, R2008 ed. 2021.

Response to reviewer 2

General Comments. *This paper presents excellent experimental results of an I-FOG using a hollow-core fiber (HCF), or air-core, for the sensing coil. Bias noise and drift come quite close to the ones obtained with a classical polarization-maintaining (PM) silica-core fiber, which is a great progress compared to previous reports with an HCF.*

Response. We sincerely thank the reviewer for the positive and encouraging comments “...presents excellent experimental results of an I-FOG using a hollow-core fiber (HCF) ...a great progress compared to previous reports with an HCF”. Your feedback reinforces the value of our research.

Comments 1. *From this standpoint, it deserves publication, however it must be with a major mandatory revision since the important analysis of the Shupe effect is uncomplete, not to say wrong!*

Response: Thank you for your valuable comments. Your feedback closely aligns with Reviewer 1's Comment 1. In the revised version of the manuscript, we have made substantial modifications to address the analysis of the Shupe effect. Specifically, we have incorporated an analysis of the Shupe constant variation in a real fiber coil under non-uniform longitudinal strain distribution. Detailed changes can be found above.

Comments 2. *As the authors say, HCF were recognized very early as very suitable for an I-FOG, since only a very small part of the light propagates in silica which avoids several spurious effects: non-linear Kerr effect (but it is solved in an I-FOG with a broadband source), backscattering (but it is not a problem either with a broadband source), magnetic dependence (but this remains manageable with μ -metal shielding). The one that remains a problem is the Shupe effect and there was great expectation of improvement in an early publication of 2007 (Ref. 18 of the paper).*

The important improvement showed in this present paper is that the attenuation is greatly reduced (0.38 dB/km at 1550 nm, whilst it is 0.5 dB/km for a commercial PM fiber) with the recent concept of anti-resonant HCF (ARF), and that an improved design developed by this team, “tDNANF”, provides in addition a good conservation of polarization, which is key for an I-FOG.

A good way to compare bias performance of I-FOGs is to normalize with the area of the sensing coil. Here, with a length $L=469$ m, a diameter $D=136$ mm, the area $A=LD/4$ is 16 m², and the ARW, i.e. the bias noise, is 0.0038 deg.h-1/2.

A typical result for a navigation-grade commercial product can be found in figure 2.22 of reference 1. ARW is 0.0013 deg.h-1/2 for an area of 25m², which is about twice better in normalized value. The difference is even larger taking a prototype of 200 m² that has

an ARW of 0.000032 deg.h-1/2, as seen in figure 9.9 of the same reference 1. This is then 10 times better in normalized value, but it uses RIN compensation technique.

One can then say that this “tDNANF” I-FOG is very close to classical I-FOGs in terms of ARW, which is excellent scientifically, but not useful business wise, since the cost of a classical PM fiber is much smaller

Response: Thanks for the valuable comments and suggestions. In alignment with your feedback and Reviewer 1’s Comment, we have replaced the term "the Shupe effect" with “the temperature and strain induced time-transience effects” to more accurately describe the complex non-reciprocal phenomena encountered in IFOGs.

Additionally, we have adopted your suggested fair comparison method for evaluating ARW performance and incorporated the specific example of a navigation-grade commercial gyro into the revised manuscript. Your insightful suggestions have substantially enhanced the quality and clarity of our paper, and we are deeply grateful for your support.

Changes in the revised manuscript

In the second paragraph of the Section “**Introduction**”, the sentence has been revised to “such as the temperature and strain induced time-transience effects^{8,9}”.

To highlight the rapid progress in attenuation reduction achieved through the concept of anti-resonant hollow-core fibers (ARF), we have included a recent publication from our team as Ref. [26], which also demonstrates a loss level of 0.1 dB/km.

In the fourth paragraph of the Section “**Results**”, Subsection “**Gyro test**”, we have added “To provide a fair comparison, we adopt a commercial navigation-grade gyro, such as the solid-core PMF gyro in Fig. 2.22 of Ref. 1, and normalize the ARW by the sensing coil area. Our ARW performance is very close to---less than twice worse than---that of classical IFOGs, which is excellent”.

Comments 3. *The main interest of this “tDNANF” I-FOG looks like the 10-fold reduction in the Shupe effect that the authors observed experimentally. To me, this result is strange, to say the least. It is known that there is some dispersion of the practical Shupe coefficient in an industrial production because of winding defects and it happens that some coils have a low Shupe coefficient. I think it is the case here.*

Response: Thanks for the valuable comments. As the reviewer correctly points out, manufacturing imperfections during fiber winding can occasionally lead to a random compensation of the Shupe effect, resulting in exceptionally favorable temperature characteristics of certain coils. However, such occurrences are rare and represent a low-

probability event. Generally, our winding process maintains consistent performance within a specific range, making the likelihood of achieving such exceptionally favorable temperature characteristics minimal.

Regarding the significant reduction in temperature sensitivity, we believe this improvement stems from simultaneous enhancements in both the pure Shupe effect and the Mohr effect. The pure Shupe effect arises from spatial and temporal drifts in the temperature field when the fiber is not subjected to stress. In contrast, the Mohr effect occurs due to spatial variations in stress distribution within the fiber combined with temporal changes in temperature, leading to phase drift.

Preliminary estimates indicate that our four-tube tDNANF exhibits a pure Shupe effect that is approximately **30 times weaker** and a Mohr effect that is about **14 times weaker** compared to conventional PMF. These estimates take into account the beneficial factor of $n_g \times n_{\text{eff}}$.

Changes in the revised manuscript

After the fourth paragraph of the Section “**Results**”, Subsection “**Fibre optical characterization**”, we have added a new paragraph to address the issue of strain-induced Shupe constant variation within a coil, specifically the Mohr effect. We divide the time-transience nonreciprocal effects into two temporal regions ($d^2T/dt^2 \neq 0$ & $dT/dt \neq 0$), and evaluate the strength of thermally induced strains by calculating fiber stiffness. A larger stiffness yields a higher resistance to Mohr effect. Please see “**However, it is widely recognized that in a real fibre coil, the Shupe constant is not uniform due to differential longitudinal strains. When the coil is in the temperature regions with a non-zero first-order time derivative ($dT/dt \neq 0$), the Mohr effect---caused by layer-to-layer variation in the Shupe constant under changing temperature---becomes the primary source of thermal insensitivity^{1,9}. To evaluate the impact of the Mohr effect, we calculate the fibre stiffness, defined as the product of Young’s modulus and cross-sectional area. Our four-tube tDNANF exhibits a 7.07-fold increase in stiffness compared to conventional PMF (see Supplementary Material S1.4). Combined with the beneficial factor of $n_{\text{geff}} \times$, the strain induced time-transient effect in our tDNANF gyro is expected to be ~ 14 times smaller than that of an SCF gyro**”.

Comments 4. *The authors must read carefully chapter 6 of their reference 1. It is explained in it that the formula (equation 1 of the paper) of the original Shupe paper is incomplete. It assumes that the Shupe coefficient is uniform along the coil which is not the case at all because of the thermal expansion of the coating. Because of leverage effects, the fiber loops cannot block the transverse expansion of the coating, despite the high rigidity of silica, which yields differential longitudinal strains between layers (figure 6.23 in particular) and this is confirmed with BOTDA, Brillouin Optical Time Delay Analysis (Figure 6.25). This differential longitudinal strain is by far the main*

temperature-dependent effect. This was measured with a classical silica core fiber, but the same differential strain happens with a hollow-core fiber! Such a differential strain yields about the same phase difference in a classical fiber or in a hollow-core fiber. It is written at the top of page 151 of reference 1. This effect needs to be described and referenced by the authors!

In addition, the pure Shupe effect on the whole coil does not follow a $T \dot{}$ (first order derivative law) even if it looks like it in equation 1; it follows a $\Delta T \dot{}$ law that is close to the second order derivative (see figures 6.5 and 6.9).

They can give their experimental result, but they must add that it should not be that low and that further investigation is needed using several coils.

To finish, the measurement of the Shupe constant in S1.2 is not using a coil, and then there is not this differential strain between layers.

Response: Thank you for these insightful comments and suggestions. We agree with the reviewer's critique and acknowledge the need for further clarification.

Changes in the revised manuscript

In the first paragraph of the Section “**Results**”, Subsection “**Thermal sensitivity**”, we have revised the text to “Combining the *pure Shupe* and *Mohr* effects, a temperature- or strain-induced phase shift occurs between the two counter-propagating beams in the fibre coil of an IFOG. This phase shift is indistinguishable from the rotation-induced phase shift. The resulting rotation rate error (Ω_E) can be expressed as

$$\Omega_E = \frac{1}{L} \left\langle \frac{\Delta S}{S} \right\rangle \dot{T} \quad (1)$$

where L and D are the total length and diameter of the coil, respectively, and $n_{\text{eff}}(n_g)$, S , and \dot{T} denote the effective (group) modal index, the Shupe constant, and the time derivative of temperature in a fibre segment of length dz at a distance z from one end of the coil. The symbol Δ stands for the difference between the two values at z and $L-2z$, while the angle brackets $\langle \rangle$ denote the mean of the values at these two sites.”

Comments 5. *Another known problem of hollow-core fibers is how to connect them. It cannot be done directly on the IO circuit because of mode size mismatch. The photos of figure 4 must be enlarged and the set-up must be better explained. It must be said that it is more complicated than the case of classical fibers even if it remains manageable.*

Response: Thanks for the valuable comments. Connecting hollow-core fibers to the IO circuit with low coupling loss is a crucial technique in our work. To address the mode size mismatch inherent in HCF, we employ two pairs of microlenses to effectively adapt

the mode size. Additionally, we utilize a microprism to translationally offset one of the collimated beams, thereby preventing crosstalk between the two beams. In response to the reviewer's suggestion, we have enlarged the photos of Figure 4 to enhance clarity. While the connection process for HCF is indeed more complex than that for classical fibers, our methodology ensures that it remains manageable and efficient.

Changes in the revised manuscript

In the first paragraph of the Section "**Results**", Subsection "**Gyro test**", we have added the phrase "**using microlens pairs for mode size adaption**" to highlight the technique used for fiber-to-chip direct coupling.

Response to reviewer 3

General Comments. *The manuscript by Prof. Ding et al. presents an exceptional and meticulously conducted study on interferometric fiber optic gyroscopes (IFOGs). This work introduces a groundbreaking navigation-grade IFOG, distinguished by its novelty and methodological rigor. By leveraging the combined strengths of a quadrupolar-wound coil of four-tube truncated double nested antiresonant nodeless fibre (tDNANF), the authors achieve remarkable advancements in fibre medium properties, including low loss, minimal bend loss, single modality, and outstanding linear polarization purity. These innovations enable the device to achieve an impressive angular random walk (ARW) of $0.0038\% \sqrt{h}$ and a bias stability (BS) drift of $0.0014\%/h$ over 8500 seconds.*

The findings hold significant potential for advancing high-precision inertial navigation applications, positioning this work as a substantial contribution to the field. The manuscript is thoughtfully organized, with a clear contextualization of the research problem, a robust discussion of the results, and well-designed figures and supplementary materials that effectively support the authors' claims. While the manuscript is highly commendable and suitable for publication in Nature Communications, a few minor revisions would enhance its clarity and accessibility to a broader audience.

Response. We greatly thank the reviewer's positive comments that “... presents an exceptional and meticulously conducted study...achieve remarkable advancements in fibre medium properties...hold significant potential for advancing high-precision inertial navigation applications, positioning this work as a substantial contribution to the field...”.

Comments 1. *The manuscript highlights that DNANF has achieved a record loss of <0.11 dB km⁻¹ at 1550 nm, surpassing all other optical fibres. For the DNANF used in this study, Figure 3a reports an average propagation attenuation of 0.38 dB km⁻¹ within the wavelength range of 1525-1565 nm. Could the authors elaborate on the factors contributing to the additional loss in the current implementation? Furthermore, how much performance improvement could be anticipated if a DNANF with a propagation loss of 0.11 dB km⁻¹ were used?*

Response. Thanks for the reviewer's comments. Indeed, 2024 has seen significant advancements in the attenuation properties of DNANF. In March, the Southampton team reported a minimum loss of <0.11 dB/km using a five-fold structure, which is unsuitable for gyro applications. More recently, our team has achieved a minimum loss of 0.1 dB/km with our four-fold tDNANF (see the newly added Reference [26] in the revised manuscript).

Regarding the average loss of 0.38 dB/km for the straight four-fold tDNANF used in this study, the additional loss may arise from the following factors:

1). Confinement loss: As discussed in Reference [26], maintaining high mode purity

necessitates enlarging the crescent-shaped air region between the truncated large tube and the middle tube. This design facilitates phase matching between the high-order modes in the core and the fundamental modes in the air crescent. However, it also results in a slight increase in confinement loss. Since high mode purity is essential for gyro applications, we prioritized it over minimizing loss.

2). *Fabrication imperfections*: Based on our experience, even with identical structural designs, the fabricated fibers exhibit variability in loss levels. This variability is likely due to fabrication imperfections, including structural nonuniformities both longitudinally and transversely, as well as cleanliness issues within the structures. A comprehensive analysis of these effects is complex, but achieving a loss of 0.38 dB/km is typical under laboratory conditions.

Regarding the performance improvement anticipated with a four-fold tDNANF having a propagation loss of 0.11 dB/km, the primary advantage is the ability to utilize longer fiber lengths within the same loss budget. We plan to focus on this aspect in our future work and evaluate its specific benefits in practical applications.

Changes in the revised manuscript

We have updated the manuscript by adding a new Reference [26], which has been accepted by *Optica* and is currently in press.

Comments 2. *Figure 3a reports a propagation attenuation of 0.38 dB km⁻¹ with the fibre wound on a drum with a radius of 16 cm, while an averaged macrobend loss of 4.7 dB km⁻¹ is observed for a fiber bent to a radius of 6 cm for 100 turns. In Fig. 3b, the linear fit shows an additional loss of 3.3 dB km⁻¹ for a fiber bent to a 6 cm inner radius and a 7.6 cm outer radius over 36×30=1080 turns. The authors state that these results are consistent with the measured macrobend loss. Could the authors explicitly describe how the losses in Figure 3b align with those reported in Figure 3a?*

Response. The macrobend loss measured in Fig. 3a (4.7 dB km⁻¹) is slightly larger than the result in Fig. 3b (3.3 dB km⁻¹), primarily due to the following two factors:

- 1) *Bending Radii Differences*: In Fig. 3a, the additional loss was measured by bending the fiber into a radius of $R_b = 6$ cm with 100 turns. In contrast, in Fig. 3b, the fiber was coiled into a loop with an inner diameter of 6 cm and an outer diameter of 7.6 cm. The latter configuration results in a larger average bending radius, which leads to a slightly lower macrobend loss.
- 2) *Coiling Methods Variations*: The coiling methods used in the two measurements differ. In Fig. 3a, the fiber was manually coiled without applying longitudinal stress. Conversely, in Fig. 3b, the fiber was coiled using the quadrupole winding method with an applied stress of a few grams. This method releases a portion of the stress imposed by the acrylate coating on the fiber silica, likely decreasing the microbend loss and contributing to the lower bend loss measured.

Considering these two factors, the lower loss observed in Fig. 3b is consistent with the result of Fig. 3a.

Changes in the revised manuscript

We have updated the manuscript accordingly. Please refer to the second paragraph of the Subsection “**Fibre optical characterization**”. Please see “An overall extra loss of 1.51 dB was detected during winding when the 469 m fibre length was wound with 36 layers and 30 turns, featuring an inner radius of 6 cm and an outer radius of 7.6 cm. The linear fit in Fig. 3b indicates an additional loss of 3.3 dB km^{-1} , which is consistent with the measured macrobend loss at $R_b = 6 \text{ cm}$. After curing, the coil yielded a total loss of 4.23 dB, comprising 2.53 dB additional loss from potting adhesive curing, 1.51 dB additional loss from winding, and 0.19 dB fibre background loss.”

Comments 3. *Based on the comparison of crossed-polarizer transmission spectra shown in Fig. S5, what might be the underlying reasons for lower transmission and a smaller PER for five-tube NANF? A brief discussion of these potential factors would enhance the reader’s understanding of the trade-offs between these fiber architectures.*

Response. We appreciate the reviewer’s insightful comments.

First, the transmission spectrum is a relative measurement primarily influenced by (1) the output intensity of the light source and (2) the coupling efficiency of the optical path. These parameters differ between the two types of fibers (four-tube ARF vs. five-tube ARF), leading to variations in the observed transmission spectra. In this study, these differences do not significantly impact our conclusions and can be considered negligible.

Secondly, the lower PER observed in the five-tube NANF is attributable to the factors discussed in Fig. 2 of the main text. As illustrated in Fig. 2, the five-tube NANF exhibits birefringence that is an order of magnitude lower (on the order of 10^{-7}). This reduced birefringence causes the principal axis angle to rotate rapidly with wavelength, particularly when the bending radius is small ($R_b = 6 \text{ cm}$ in this measurement). In Supplementary Material S2.1, we present a model demonstrating that increased rotation of the polarization axis leads to a significant decrease in the PER of the fiber.

Comments 4. *While the presented DNANF-based IFOG demonstrates navigation-grade performance, what specific advancements or optimizations would be required to achieve a strategic-grade IFOG? For example, could improvements in fiber fabrication, such as reducing propagation loss further or minimizing environmental sensitivity, play a critical role? Additionally, are there innovations in coil design, signal processing, or thermal management that could bridge this gap? Including a brief forward-looking perspective on the pathway to achieving strategic-grade performance would significantly enrich the manuscript.*

Response. We thank the reviewer for the valuable suggestion. Based on conventional

solid-core fibers, strategic-grade IFOGs have achieved commercial implementations. During this technological development, the industrial community has accumulated numerous advancements in circuit technology, light source technology, and RIN noise reduction techniques. For air-core fibers aiming toward strategic-grade IFOGs, these advanced technologies will, of course, play vital roles. Additionally, replacing the fiber core medium with air could potentially bring further performance enhancements by effectively eliminating detrimental effects such as Kerr and stimulated Brillouin scattering (SBS). This would enhance the signal-to-noise ratio and reduce the ARW. We believe that the key component for a strategic-grade air-core IFOG will be a HCF with a long, intact length and excellent comprehensive optical properties, including low loss, low bend loss, high mode purity, polarization maintenance, and appropriate sizing. The specific techniques developed in this study, such as lossless winding and miniature, low-loss fiber-to-chip coupling, pave the way toward achieving this goal.

Regarding the *“forward-looking perspective on the pathway to achieving strategic-grade performance”*, we maintain a humble outlook and will continue our research and development efforts to realize these technological advancements.

Re: ID: NCOMMS-24-41217B

Title: Navigation-grade interferometric air-core antiresonant fibre optic gyroscope with enhanced thermal stability

Authors: Maochun Li^{1,†}, Yizhi Sun^{2,3,5†}, Shoufei Gao^{2,3,5†}, Xiaoming Zhao^{1,*}, Fei Hui¹, Wei Luo¹, Qingbo Hu^{2,3}, Hao Chen^{2,3}, Helin Wu^{2,3}, Yingying Wang^{2,3,5}, Miao Yan^{1,4}, and Wei Ding^{2,3,5,6*}

Response to reviewer 1

General Comments: *I thank the authors for their detailed responses and for their having better illuminated this important and interesting work.*

Response. We thank the reviewer's kind comments and high evaluation.

Comments 1. *There is one final point that I believe must be clarified in the final manuscript. It is unclear how the new 40 nm PER of 20.4 dB was measured in the gyro coil. If this was measured by fitting an equivalent birefringence alignment angle and then calculating an equivalent crossed polarizer PER, then this should be clearly stated. For this calculation method, the magnitude of the "empirical factor" (Δ in eq. 2) that accounts for polarization coupling through the coil should also be reported since this empirical factor represents "lossy interpolarization coupling between sequential segments along the fibre", i.e. precisely the quantity of concern in a PER measurement. Ideally, the measured crossed-polarizer spectra should also be reproduced in the manuscript or supplementary material to avoid all doubt.*

On the other hand if this coil PER was measured in the conventional way, e.g. by taking the ratio of total broadband throughput power measured at two half-wave plate orientations, then this should be stated explicitly in the manuscript, but no further support is required.

Response. Thanks for the reviewer's comments. The PER of the 469m coil was determined using a commercial PER meter (Fiberpro, ER3100) and an ASE broadband light source. It is a routine testing procedure. To avoid confusion, we add a description of the PER meter in the caption of Fig. 3f in the main text. Please see "...by using an ASE source of 40 nm bandwidth and a commercial PER meter (red star).".

Comments 2. *Apart from this, the revised manuscript and authors' response have satisfactorily addressed all my concerns.*

Response to reviewer 2

General Comments. *The second submission of this paper presents even better experimental results of an I-FOG using a hollow-core fiber (HCF), or air-core, for the sensing coil. Bias noise and drift come quite close to the ones obtained with a classical polarization-maintaining (PM) silica-core fiber, which is a great progress compared to previous reports with an HCF.*

The new bias test and the resulting Allan deviation, now over 185 hours without seeable bias instability (instead of 24 hours in the first submission), show clearly that it is a very promising approach.

From this standpoint, it clearly deserves publication.

Response. We sincerely thank the reviewer for the insightful and precise comment on our work. We deeply appreciate the recognition of the progress made in our experiments.

Comments 1. *The main concern I had in my first review, was the analysis of the Shupe effect that was uncomplete and did not consider the Mohr effect. It is clear that the pure Shupe effect is greatly reduced, but to me, for the Mohr effect, the new argument of the authors about the stiffness of their fiber is not convincing at all (end of 1st column of page 5). They say that the stiffness is increased by a factor 7 ... but it is because of the thick outer silica cladding (figure S.4) and not because of the hollow core structure that has a very small stiffness! Figure S.4 must be in the paper and not in the supplementary materials.*

One may say, what happens with a solid core fiber with the same outer cladding diameter (238 μm)?

To me, there is still Mohr effect, even if I am not sure to be right.

Response: We sincerely thank the reviewer for very insightful analysis and helpful suggestions. We agree that the increase in stiffness of our large-diameter fiber likely contributes to the 7-fold reduction in the Mohr effect, while the remaining 2-fold reduction may stem from the decrease in $n_g \times n_{\text{eff}}$ in the air-core fiber compared to conventional solid-core fibers. This suggests that, even with the same outer diameter, an air-core IFOG may experience a reduced Mohr effect compared to a solid-core IFOG. Of course, further experimental validation is required to confirm this concept.

Regarding the definition and calculation of the fiber's stiffness (Figure S4), we recommend that readers refer to Ref. 32 for a detailed explanation. To clarify this in the paper, we have added a superscript citation to Ref. 32 in the fifth paragraph of the

“Fibre optical characterization” subsection, under the **“Results”** section.

Comments 2. *However, let’s converge! We are not going to continue to discuss this submission many times. And the bias result is by itself worth publishing.*

Response: Thanks for the reviewer’s support.

Comments 3. *I propose that the authors put in their conclusion:*

“Excellent results in terms of bias noise and drift for an air-core fiber. We have also observed a very good reduction of the Mohr effect, but it will be needed, in a future paper, to have further measurements with other coils to valid this reduction of the Mohr effect, since earlier publications think that it should be about the same as with solid-core fibers.”

Response: Thanks for the valuable suggestions. Regarding the recommended first sentence, we believe our previous draft has already clearly addressed it. Specifically, we mention: **“With an ARW of 0.00383 deg h^{-1/2} and a BI drift of 0.0017 deg h⁻¹...”**. For the second suggestion, we have made revisions by incorporating the phrase **“...has been experimentally observed, thanks to the optimization of both the pure Shupe and Mohr effects, ...”** in the fifth paragraph of the **“Discussion”** section. Additionally, we have added the sentence **“To further validate this reduction in the Mohr effect, future measurements with other coils will be needed.”** in the fifth paragraph of the **“Fibre optical characterization”** subsection, under the **“Results”** section.

Response to reviewer 3

General Comments. *I have thoroughly reviewed the revised manuscript and the accompanying response letter and found that all the concerns raised in the previous round of review have been adequately addressed. I believe the revised manuscript is now suitable for publication in Nature Communications without any further changes.*

Response. We sincerely thank the reviewer's support.